# Sub-diffraction error mapping for localisation microscopy images

Richard J. Marsh[1], Ishan Costello[1], Mark-Alexander Gorey[1], Donghan Ma [2], Fang Huang [2], Mathias Gautel [1], Maddy Parsons [1] & Susan Cox [1✉]

Assessing the quality of localisation microscopy images is highly challenging due to the difficulty in reliably detecting errors in experimental data. The most common failure modes are the biases and errors produced by the localisation algorithm when there is emitter overlap. Also known as the high density or crowded field condition, significant emitter overlap is normally unavoidable in live cell imaging. Here we use Haar wavelet kernel analysis (HAWK), a localisation microscopy data analysis method which is known to produce results without bias, to generate a reference image. This enables mapping and quantification of reconstruction bias and artefacts common in all but low emitter density data. By avoiding comparisons involving intensity information, we can map structural artefacts in a way that is not adversely influenced by nonlinearity in the localisation algorithm. The HAWK Method for the Assessment of Nanoscopy (HAWKMAN) is a general approach which allows for the reliability of localisation information to be assessed.

[1] Randall Centre for Cell & Molecular Biophysics, Guy's Campus, King's College London, London, UK. [2] Weldon School of Biomedical Engineering, Purdue University, West Lafayette, IN, USA. ✉email: susan.cox@kcl.ac.uk

Image assessment and validation are critical for all microscopy methods, to ensure that the image produced accurately reflects the structure of the sample. When image processing is an integral part of the technique, as in single-molecule localisation microscopy (SMLM) methods[1,2], it is important to verify that the image processing is not producing errors or biases. A number of different types of artefact can be introduced by SMLM image processing, including missing/biased structure, blurring and artificial sharpening. These artefacts occur when emitter fluorescence profiles overlap in the raw data and are incorrectly localised towards their mutual centre, introducing a bias that is often substantial when compared with the estimated localisation precision[3–7]. The frequency and scale of localisation bias varies greatly with the local structure and the emitter density. Here we use the term 'artificial sharpening' to mean any reconstruction artefact that results from localisation bias, as biased localisations generally show substantially reduced scatter compared to unbiased ones as the emitter density increases.

The importance of assessing the quality of SMLM images is widely recognised but is a challenging problem to solve. In general, images must be assessed without access to the ground truth structure, meaning that any image assessment method must make some type of comparison with alternative data or analysis. For example, Fourier ring correlation (FRC)[8] splits the dataset into two images, both of which will be subject to the same algorithmic bias. Therefore, when artificial sharpening is present, FRC will simply report the reduced scatter of localisations typical of biased reconstructions as higher resolution. Most localisation algorithms suffer from similar artificial sharpening effects, so comparisons made between them have limited effectiveness[5]. Here, we use the term resolution to indicate the length scale at which resolvable structures are authentically reproduced in the reconstruction. In other words, the local resolution is the scale at which the measured biases are smaller than the scale of the structures themselves.

This problem of common biases in each image is averted in the super-resolution quantitative image rating and reporting of error locations method (SQUIRREL)[9]. This method downscales the SMLM image and compares it to a linear transformation of the widefield image. However, this has two major disadvantages. Firstly, the downscaling eliminates the fine structure in the image, meaning that only differences on or above the scale of the PSF can be quantified (see Supplementary Note 1 and Supplementary Figs. 1, 2). Secondly, sharpened images will score more highly than accurate reconstructions if their reconstruction intensity has a more linear relation to their labelling density (which is likely for a substantial number of algorithms, see Supplementary Fig. 2).

Several factors contribute substantially (and generally) to nonlinearity in SMLM reconstructions, including the background fluorescence and the degree of sampling. Particularly when illuminating in HILO mode, the background signal arises from fluorophores at a different z position to the focal plane. They are therefore exposed to a different intensity of light and will have substantially different blinking properties which can make them less likely to be detected. Conversely, if the background is close to focus, then the much higher labelling density in the sample structure may cause a higher proportion of background fluorophore to be detected compared to the foreground (as some foreground localisations are missed due to high emitter density). SQUIRREL's optimisation process assumes all variation in the reference image is reflected in the density of localisations in the reconstruction, meaning that nonlinearity will be reported as errors.

With regard to sampling, the intensity in a widefield image (where molecules don't blink) would normally be an accurate reflection of the labelling density. However, in a typical SMLM experiment, emitters can make multiple appearances (with different intensities) or not at all, meaning that the reconstruction intensity is not reliably related to the number of local fluorophores. This limitation can be circumvented if the sum of the acquired localisation microscopy frames is used as the widefield reference, as this assures the sampling is the same as in the super-resolution measurement. However, this eliminates the ability to detect missing (non-sampled) structure in the test image (although under sampling due to overlapping emitters could still in principle be detected.). To avoid these difficulties, we use a second super-resolution (HAWK preprocessed) image as the reference and both images are binarised to remove the dependence on intensity and sampling information (see Supplementary Note 1 for a further discussion of these effects).

There are also approaches to quantify how the algorithm used for SMLM can limit accuracy/resolution and introduce bias[5,10]. While these can demonstrate bias and artificial sharpening when the ground truth structure (or some defining property e.g. as with a spatially random structure) is known[5,10], and can be used to assess the relative performance of algorithms, they cannot assess the quality of reconstructed experimental images. Additionally, the relative performance of different algorithms on simulated test data cannot be guaranteed to reproduce the effects observed on real samples containing varied types of structure.

Here we introduce an alternative approach which allows visual quantification of the accuracy of an SMLM algorithm's reconstruction. Our method uses HAWK[11], a preprocessing step that for any particular algorithm eliminates the artificial sharpening caused by excessive density in the raw data, at the cost of a small decrease in localisation precision. This was demonstrated using ThunderSTORM[12], a standard fitting algorithm that can perform single emitter (SE) and multi-emitter (ME) fitting. HAWK has also been demonstrated to decrease bias, artificial sharpening and nonlinearities in reconstruction intensity when used with other algorithms.

## Results

**The HAWKMAN algorithm.** We exploit the accuracy and reliability of HAWK to identify potential artefacts in a localisation microscopy image produced without HAWK, and to indicate where HAWK preprocessing has reduced localisation precision sufficiently that underlying fine structure could have been made unresolvable. This is achieved by quantifying structural differences between the original image and the HAWK-processed reconstruction as produced using the same algorithm. This measure can be used to map out areas which have artefacts, such as artificial sharpening, and to ascribe a confidence level to local regions of the image at the sub-diffraction level. A measure of the local resolution (in the sense defined above of the length scale at which structure is correctly reproduced), can be ascertained by progressively blurring the input images with a Gaussian kernel for longer length scales and repeating the comparison. The length scale at which reasonable agreement (quantified by the degree of local correlation) between the HAWK-preprocessed and non-HAWK-processed output images in the local region is achieved, indicates the resolution obtained. From this, a map of the maximum scale of artefacts in the image can be produced.

HAWKMAN takes as input data a test (super-resolved reconstructed) image and reference (HAWK-preprocessed) image, and a maximum length scale over which the performance of the algorithm will be evaluated (Fig. 1). The input images are intensity-flattened to suppress the influence of isolated outlying high intensity points (due to repeated sampling), which frequently occur in SMLM. The flattened images are then blurred with a Gaussian kernel of width equivalent to the current length

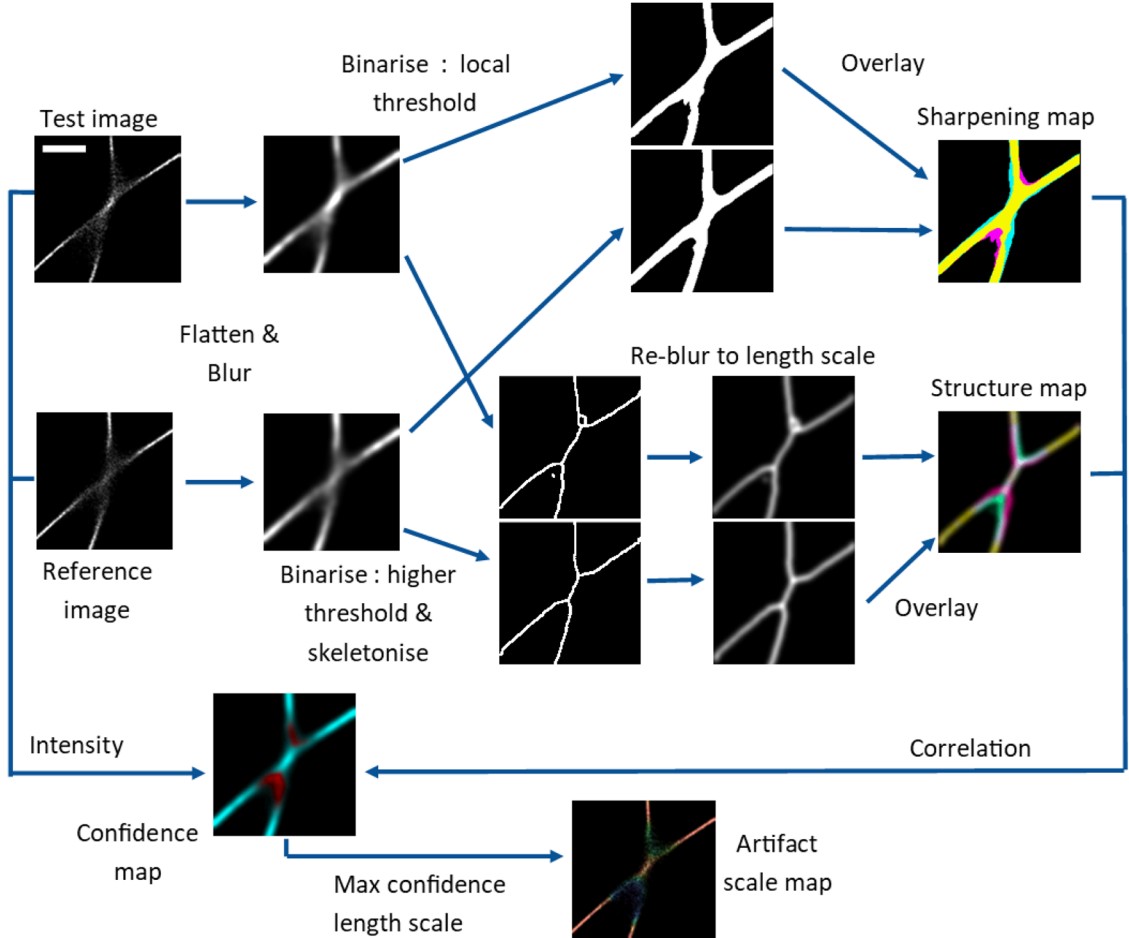

**Fig. 1 HAWKMAN method and results on experimental data from the Localisation Microscopy Challenge.** The processing steps of the HAWKMAN algorithm, producing the sharpening map, structure map and confidence map. For the sharpening and structure maps areas of input image agreement are displayed in yellow whereas magenta and cyan indicate structure only present in the test and reference images, respectively. Therefore, magenta indicates artificial sharpening and cyan indicates missing structure. The confidence map indicates the calculated local correlation score (see Methods for details), ranging from 0 (red) to 1 (cyan), whilst brightness denotes the intensity of the original images Repeating the process at many length scales can indicate how the size of artefacts varies over the structure as indicated in the artefact scale map. Scale bar 500 nm.

scale of interest (ranging in integer multiples of a single pixel up to a user-specified maximum). These blurred images are then binarised according to a length scale-specific adaptive threshold. This produces two images for each input: one (the sharpening map) is produced from a threshold of roughly 50% of the local maximum, the other (the structure map) is produced from a higher threshold (85%) and subsequently skeletonised and then re-blurred with the same Gaussian kernel. It should be noted that the optimum thresholds will vary slightly depending on the local dimensionality of the sample, but these variations are not critically important. A higher threshold may be required if there is only a small difference in the labelling density of adjoining structure, such as may result from nonspecific labelling. The optimum value is only required to resolve these adjacent structures, default parameters will still identify strong sharpening artefacts. Supplementary Fig. 3 shows how to choose the optimum parameters. Default parameters are used for all data presented here except for the clathrin-coated pit data below.

Differences highlighted in the sharpening and structure maps indicate areas of artificial sharpening, bias and/or large differences in precision. The local correlation between them is used as a confidence metric for the reliability of that region in the test reconstruction, giving the confidence map (Fig. 1). The procedure is repeated for increasing length scales up to the

maximum of interest (typically the instrument PSF). This allows the accuracy of the localisation algorithm to be assessed across the whole range of length scales. This approach bears some similarities with the Gaussian scale-space technique[13], that has been used in the fields of image processing and computer vision. In this technique, the input image is also examined simultaneously at different scales buy convolution with Gaussian kernels of different sizes. Images can then be further processed on structure only of a specific scale. Our approach is to compare only the structure information in the test and reference input reconstructions at each scale to detect which scales contain disagreement and therefore density induced artefacts.

As an optional further stage, a map of the largest scale artefact identified at each point in the image can be produced from the confidence map. This, the artefact scale map, consists of the combined input image colourised to reflect the smallest length scale that that part of the image has high reported confidence. This enables a rapid appraisal of which parts of the reconstruction contain acceptably accurate structure and which do not. Although the artefact scale map provides a useful summary of the HAWKMAN output. The other maps better indicate the magnitude of artefacts at each scale and the confidence map gives an indication of how the structure would appear at a resolution known to only contain accurate information.

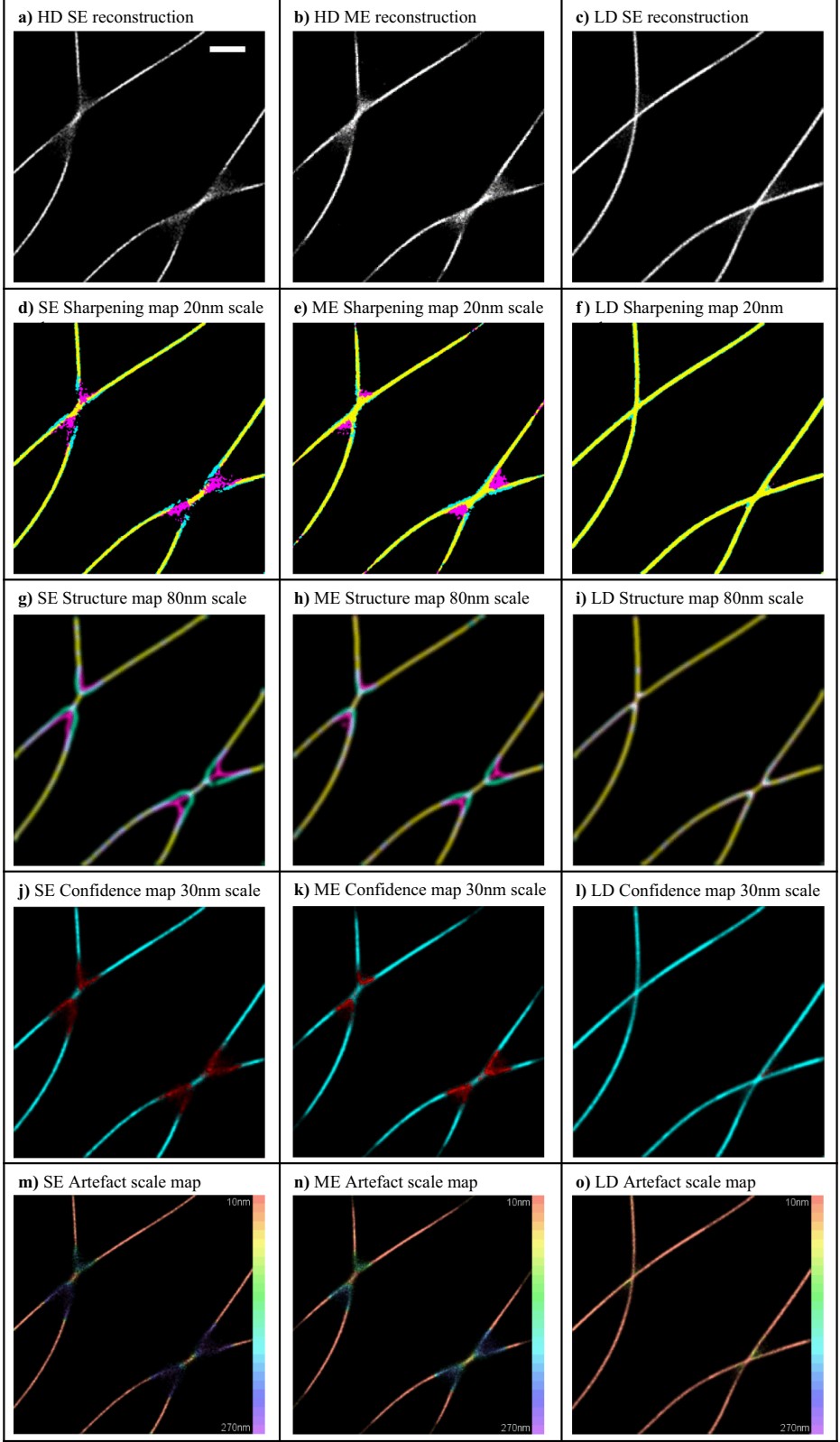

**Testing with benchmark data**. In order to test the performance of HAWKMAN we selected simulated datasets from the Localisation Microscopy Challenge[14,15] where both low and high emitter density datasets were available. This allowed us to benchmark the performance of HAWKMAN when using a HAWK reference image, against when using a low density (ground truth) reference image, which ensured that the location and scale of artefacts in the test image was known. Figure 2 shows the results of HAWKMAN analysis on simulations of microtubules for the Localisation Microscopy Challenge for both SE and ME Gaussian fitting of the data. For areas of microtubule crossover (Fig. 2a–c) the high density reconstructions (Fig. 2a, b) show substantial sharpening artefacts not present in the reconstruction from low density data (Fig. 2c). As expected, the SE

**Fig. 2 HAWKMAN analysis of simulated microtubule data from the Localisation Microscopy Challenge.** Comparison of SE and ME fitting of high emitter density data and equivalent low density data. A section of the reconstructions is shown in (**a–c**) for HD:SE fitting, HD:ME fitting and LD:SE fitting, respectively, which contain artificial sharpening for both high density methods. The results of HAWKMAN analysis are shown in (**d–l**) for each of the three test reconstructions with HAWK used as the reference. The sharpening maps (**d–f**) and structure maps (**g–i**) are shown at a length scale of 20 and 80 nm, respectively. In both the SE and ME cases the magenta areas (test image only) in the sharpening maps (**d**, **e**) and the structure maps (**g**, **h**) indicate substantial sharpening along with missing structure (cyan), which is more severe (as expected) in the SE case. Both these effects are absent for the low density reconstruction (**f**, **i**). These results are reflected in the confidence maps (**j–l**, 30 nm scale). These show substantially reduced confidence (strong artefacts, highlighted in red) for both high density methods (**j**, **k**), whereas for the low density data the confidence (**l**) is high everywhere. The differences in accuracy of the three reconstructions is further highlighted by the corresponding artefact scale maps (**m–o**). These show the maximum scale of artefacts is larger for the SE reconstruction than the ME and that the LD is virtually artefact free. The colour scale ranges from length scales 1 reconstruction pixel (10 nm red) to 27 (270 nm purple). Scale bar 500 nm.

reconstruction (Fig. 2a) is more severely affected than the ME (Fig. 2b).

HAWKMAN analysis was performed on the challenge data sets, with the reference image being produced by the same algorithm on the same data but with prior HAWK preprocessing. Sharpening maps (Fig. 2d–f), structure maps (Fig. 2g–i) and confidence maps (Fig. 2j–l) are shown for the same length scales for each test image. The length scales in this figure (and all subsequent ones) were selected to give a clear and concise representation of the full HAWKMAN output (see Supplementary Videos 1–5 for the full sets of HAWKMAN output for each result). A more detailed description of how to use each length scale is given in the methods section but essentially as the length scale of the comparison is increased less disagreement between the test and reference images will be observed. The structure and sharpening maps indicate areas of the reconstruction that contain artefacts larger in size than the length scale examined. The confidence map quantifies the discrepancies with areas of full confidence (score = 1, coloured cyan) indicates HAWKMAN detects no artefacts larger in size than the current length scale. By examining all length scales the user can ascertain the maximum scale of artefacts in every region of the sample. The sharpening and structure maps indicate to what degree these result from sharpening or structural bias.

For the high density reconstructions, both the sharpening and structure maps show substantial biases in the microtubule positions, indicated by the magenta (test image only) where the microtubules cross. Cyan (reference image only) areas indicate some missing structure for the high density images not present in the low density reconstruction. The structure maps indicate these persist at even quite large (80 nm) length scales. This 'pinching in' of crossing microtubules is a commonly observed artefact in high density data[4,7,9,11]. The estimated degree of error is quantified in the confidence maps (low confidence marked in red, high confidence in cyan) for both SE (Fig. 2j) and ME (Fig. 2k) methods. These indicate ME fitting is more accurate than the SE fitting, but still produces substantial errors. The corresponding confidence map for the same structure simulated at low emitter density (Fig. 2l) correctly indicates no sharpening artefacts are present at this length scale (30 nm). The artefact scale maps (Fig. 2m–o) summarise the HAWKMAN output by showing the largest artefacts present at each point of the image. These show that both high density reconstructions contain large artefacts of a scale approaching the PSF near the crossing points whereas even the largest artefacts in the low density reconstruction are much smaller in scale.

The Localisation Microscopy Challenge datasets also contain experimental data, and we tested the performance of HAWKMAN on high density experimental microtubule datasets[14,15]. These were analysed using both SE & ME Gaussian fitting[12], giving results with what appear to be common artefacts (Fig. 3a, b) in the regions of clustering and microtubule crossover

including blurring, artificial sharpening and unresolved/collapsed structure. The corresponding reconstruction when HAWK is used (Fig. 3c, d) show a much more clearly resolved structure in these regions, though some precision is lost. The output of HAWKMAN analysis indicates that these are highly sharpened reconstructions. The sharpening maps (Fig. 3e, f), structure maps (Fig. 3g, h) and confidence maps (Fig. 3i, j) indicate the presence of severe sharpening artefacts even at quite large length scales (60, 160 and 100 nm, respectively).

Examining the reconstruction produced by ThunderSTORM ME fitting[12] (Fig. 3b), it can easily be seen that this is a superior reconstruction to SE fitting (Fig. 3a), but some structures are still not clearly resolved. The confidence map produced by HAWKMAN analysis for this reconstruction is shown in Fig. 3i (for the same 100 nm scale). The map shows much higher confidence in the overall accuracy of this reconstruction at this scale, yet still highlights some areas where sharpening is present. At other scales, (Supplementary Figs. 4, 5 and Supplementary Movies 1, 2) and with other high density analysis algorithms, SRRF and SOFI[16,17], (Supplementary Fig. 6 and Supplementary Movies 3, 4, 5), HAWKMAN clearly demonstrated the presence and scale of artificial sharpening and other artefacts. The maximum size of artefacts at each point in the input image is shown in the artefact scale maps (Fig. 3k, l) ranging from length scale 1 (10 nm red) to 13 (260 nm) showing these are larger in the SE reconstruction. It should be noted that for some localisation algorithms, the output reconstruction can vary significantly with the choice of parameter values used for the analysis[11,16]. Supplementary Fig. 6 demonstrates, with the example of SRRF, how HAWKMAN may aid in the optimisation of input parameters. The relative fidelity of reconstructions using different parameter sets can be assessed by using HAWKMAN on each of them.

Tests on other simulated data, also from the Localisation Microscopy Challenge, demonstrate that HAWKMAN is clearly able to detect and even quantify sharpening and other artefacts. This is the case even at length scales substantially below the diffraction limit (see Supplementary Fig. 7), right down to the scale of the localisation precision. The advantage of using a HAWK image as the reference is that it does not contain the same bias common to many other algorithms. Using a HAWK-processed high density dataset as a reference image provides comparable results to using a low density (i.e. unsharpened/artefact free) dataset, in contrast to when other algorithms are used for the reference image (see Supplementary Fig. 8). This confirms the suitability of HAWK processing for a qualitative and quantitative assessment.

**Performance comparison to SQUIRREL.** The problem of using an intensity-based comparison, as SQUIRREL does (between a measured widefield and the sampling-dependent labelling density in the reconstruction), is clearly demonstrated by comparing it with HAWKMAN. Even at the PSF scale, HAWKMAN is able to

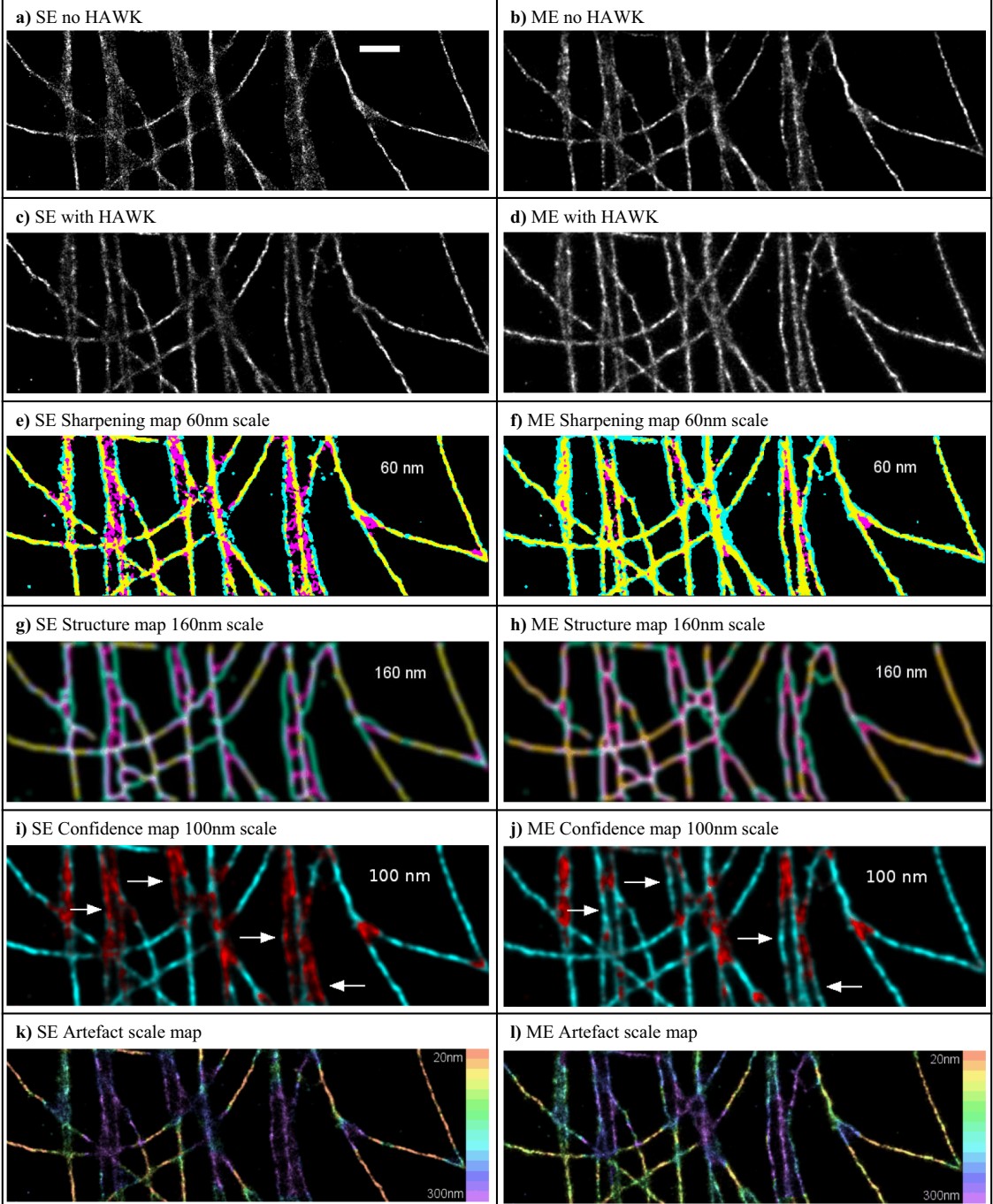

**Fig. 3 Results of HAWKMAN analysis on experimental microtubule data from the Localisation Microscopy Challenge comparing SE and ME fitting.**
The reconstructions for SE and ME fitting without HAWK are shown in **a** and **b**, respectively along with their equivalents **c** and **d** when HAWK is used. The results of HAWKMAN analysis are shown in the sharpening maps (**e**, **f**), the structure maps (**g**, **h**) and the confidence maps (**i**, **j**) for selected length scales. The SE reconstruction (**a**) shows substantial density-induced sharpening artefacts. The differences in structure between this reconstruction and the results of applying HAWK with SE fitting are highlighted by the sharpening map (**e**) shown at an example scale of 3 reconstruction pixels (60 nm using 20 nm size pixels). Substantial bias in the SE reconstruction where microtubules are in close proximity/crossing is highlighted in the structure map (**g**) even at a scale of 160 nm. Reliable regions in the reconstruction where both methods agree are highlighted in the confidence map (**i**), shown at the 100 nm scale. ME fitting (**b**) gives a more accurate reconstruction, however, as can be seen in (**f**, **h**, **j**) it still contains numerous artefacts. The confidence map (100 nm scale) (**j**) for this reconstruction shows a reduction in the severity of artefacts but some remain. The white arrows highlight some of the structure accurately reported (at this length scale) in the ME result (**j**) that were inaccurate in the SE case (**i**). A summary of the HAWKMAN results at all length scales is shown in the artefact scale maps (**k**, **l**) which indicate the maximum scale of artefacts at each area of the sample (colour indicates length scale in pixels (20–300 nm). Comparison of the SE (**k**) and ME (**l**) results shows that for ME fitting the maximum scale of artefacts is reduced in areas of high microtubule density. Scale bar 1 μm.

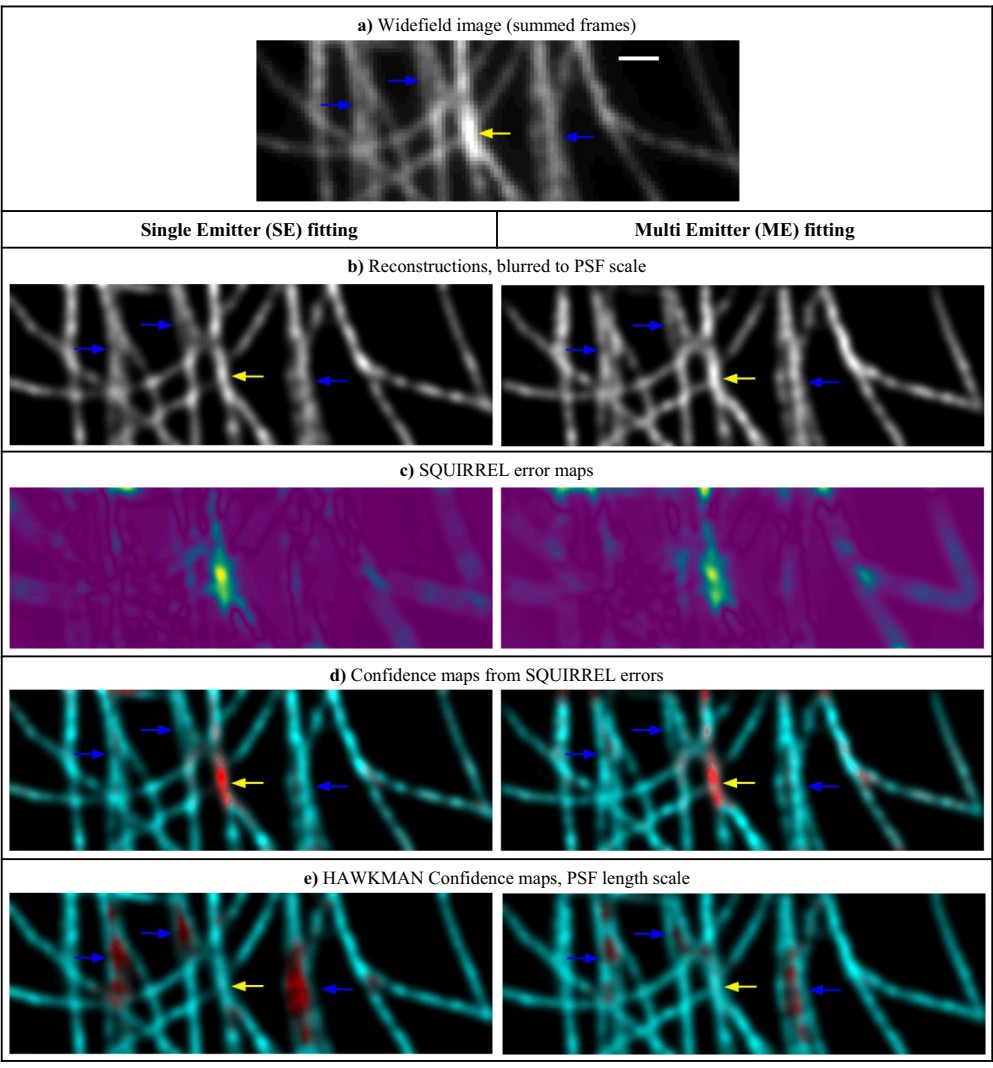

**Fig. 4 Comparison of the ability of SQUIRREL and HAWKMAN to detect structural artefacts when faced with intensity differences in the input and reference images, on experimental high density data from the Localisation Microscopy Challenge. a** A pseudo widefield image produced by summing image frames. **b** SE and ME reconstructions. Blue arrows indicate areas of obvious sharpening compared to the widefield, the yellow arrow denotes an area of high intensity. **c** Error maps produce by SQUIRREL analysis show the largest errors (yellow) are dominated by the high intensity region in (**a**). The relative error is used instead of the confidence score to produce a HAWKMAN style confidence map (**d**) from the SQUIRREL data. When compared to the actual HAWKMAN result (**e**), it shows SQUIRREL principally detects the intensity differences and does not recognise the superior structural details (blue arrows) of the ME reconstruction. Scale bar 1 μm.

correctly identify the relative fidelity of reconstructions differently affected by sharpening and other artefacts in situations where SQUIRREL cannot (Fig. 4 and Table 1). Here assessment of experimental microtubule data from the Localisation Microscopy Challenge is performed using both HAWKMAN and SQUIRREL for comparison. A widefield image produced by summing image frames (Fig. 4a) shows areas where microtubules are poorly resolved (even compared to the widefield—blue arrows) and a high intensity area (yellow arrow). The reconstruction produced by SE and ME fitting are shown in Fig. 4b, where a Gaussian blur of a scale equivalent to the PSF has been applied (the scale at which SQUIRREL makes its comparison). Comparing these with the widefield image shows significant differences, particularly in the intensity (a proxy for labelling density). The error maps produced by SQUIRREL are displayed below these (Fig. 4c). They show the largest errors are reported where there are large intensity differences between the reconstructions and the widefield reference. For easier comparison with the HAWKMAN results the relative error in the SQUIRREL maps was converted

into a confidence score (see Methods) and used to colourise the blurred reconstructions (Fig. 4d) in the same manner as for HAWKMAN. Comparison with the HAWKMAN results at the PSF length scale (Fig. 4e) highlights how SQUIRREL has detected the intensity differences (yellow arrow), but largely ignored the differences in structure (blue arrows). The converse is true for HAWKMAN, which indicates low confidence in areas of structural dissimilarity, and high confidence in the areas that differ only in intensity. HAWKMAN also reports the ME result as a much more authentic reconstruction than that from SE fitting, something that can be confirmed by inspection of the areas marked by blue arrows. Conversely, SQUIRREL rates these similarly, as any errors arising from structural differences are swamped by the much larger intensity error common to both reconstructions. This undesirable behaviour arises from the assumption that the density of localisations identified by the localisation algorithm and the pixel intensity contained in the widefield image are both linearly related to the local labelling density. This is seldom the case however due to repeat

**Table 1 Comparison of reconstruction quality metrics for SQUIRREL and HAWKMAN on challenge microtubule data.**

| Reconstruction | SQUIRREL | | HAWKMAN | |
| --- | --- | --- | --- | --- |
| | RSPCC | RSE | PCC sharpening | PCC structure |
| Single emitter | 0.911 | 68.99 | 0.834 | 0.825 |
| Multi emitter | 0.911 | 69.07 | 0.841 | 0.897 |

The table displays the quality metrics output by the two methods on the high density data analysed in Fig. 4. For SQUIRREL both the resolution scaled Pearson correlation coefficient and the resolution scaled error are unable to determine which is the better reconstruction (ME actually has a marginally higher error). However, for HAWKMAN both the correlation between sharpening maps and structure maps is significantly higher for the ME result, even at the PSF scale, indicating this is the higher fidelity reconstruction.

appearances of some emitters and nonappearance of others. It is particularly untrue with high density imaging as localisation algorithms generally progressively miss individual emitters as the density increases. Binarising the images and using the same localisation algorithm and reconstruction method for both images circumvents this problem.

Table 1 shows the quality metrics produces by SQUIRREL for the above results. These measures the correlation (resolution scaled Pearson correlation coefficient) and error (resolution scaled error) between the downscaled super-resolution image and the widefield. These are compared with analogous quantifications for HAWKMAN, the correlation coefficients between the binarised test and reference images in the sharpening and structure maps (PCC Sharpening and PCC Structure, respectively). These show that SQUIRREL is unable to detect that the ME reconstruction is superior to the SE case, as its errors are dominated by the same intensity difference. The HAWKMAN correlation scores, along with the confidence maps above clearly indicate a much more accurate reconstruction with ME fitting as expected.

Using HAWK with many algorithms does introduce a small decrease in localisation precision compared to when that algorithm is used alone[11]. The magnitude of which typically increases with the severity of bias present in the reconstruction without HAWK. For any given localisation algorithm, as the emitter density increases, the precision of the HAWK reconstruction will be reduced but will remain unsharpened. Although this leads to a reduction in the resolution attained in the HAWK image, the improvement in fidelity over the reconstruction without HAWK will increase. This means that when compared with an image of the ground truth (for instance, a low density reconstruction), using the HAWK-reconstruction as the reference image slightly exaggerates differences with the input image. This may lead to a slight overestimation of errors in some instances.

Additionally, large differences in precision between the HAWK and non-HAWK images may indicate the emitter density has exceeded the capabilities of the algorithm, even when used in combination with HAWK, resulting in reduced resolution. HAWKMAN can indicate where in a reconstruction and at what length scale the precision of the HAWK reference is sufficiently reduced that it may conceal unresolved finer structure. The experimenter is thus warned of this situation and may decide a lower emitter density is required. Alternatively, a higher performance more computationally intensive localisation algorithm may be required, such as switching from SE to ME fitting.

A demonstration of how this effect can enable HAWKMAN to detect that there may be missing fine structure, even in the HAWK reference image, is demonstrated using simulation (Supplementary Figs. 9, 10). For a limited range of emitter densities, the structure can be unresolvable but comparable in

scale to the resolution in the HAWK image. Thus, even though the structure is not visible in the HAWKed or non-HAWKed images, HAWKMAN may be able to give a first-order approximation to the underlying structure. This effect is demonstrated on the Z1Z2 domain in sarcomeres imaged at high density (see Supplementary Fig. 11 for simulated data and Supplementary Fig. 12 for experimental data). Here the structure is not visible in either input image but is present in the HAWKMAN confidence map at a 50–70 nm length scale, which is in line with the expected observation for this structure.

**Common biological structures**. To further assess the performance of HAWKMAN on experimental data we selected three well-known biological structures of differing geometry and scale to demonstrate the performance of HAWKMAN on experimental data: clathrin-coated pits, microtubules and mitochondria. This allowed us to test the performance of the system on small structures a few hundred nanometres in size (Fig. 5a–d), extended linear structures (Fig. 5e–h), and structures with an extended fluorophore distribution (Fig. 5i–l). Differences between SE-fitted data processed without (Fig. 5a, e, i) and with HAWK (Fig. 5b, f, j) are visually clear and are reflected by the HAWKMAN sharpening and confidence maps. These are displayed at representative length scales from the full HAWKMAN set for each dataset (Fig. 5) that show the presence of artefacts at a scale both a few and several times the localisation precision for each structure. As mentioned previously, a modified value of the threshold coefficient may be required for optimum performance when analysing low-contrast structures; this was the case for the clathrin data (see Methods and Supplementary Figs. 3, 13 for details). For this data the reconstruction intensity of the central region relative to their parameters range from ca. 50–75%, requiring the threshold coefficient to be raised to 0.9 in order to separate these structures, a value predictable from Supplementary Fig. 3. This change is only required to identify the artefact as being the disappearance of the 'hole' in the pits but is not necessary for HAWKMAN to detect them as artefactual in the first place, as the sharpening map will still indicate the different sizes of the sharpened structures in the test and reference images.

## Discussion

Experimentally, localisation artefacts such as artificial sharpening are particularly likely to be experienced in multicolour dSTORM measurements due to the variable performance of dyes. While Alexa Fluor® 647, with its high-intensity, low density emissions is exceptionally well-suited to dSTORM acquisitions, most other dyes are far less so. Researchers looking to perform multicolour dSTORM analysis must therefore contend with the fact that dyes whose emission spectra peak in other laser channels blink with much lower brightness and at much higher density, making them highly prone to the generation of sharpened/biased data (a situation also likely to arise when imaging living cells). HAWKMAN can identify the regions in which artificial sharpening is occurring in typical two-colour dSTORM data (see Supplementary Fig. 14). Two-colour experiments are an example of where the blinking statistics may be quite far from optimal, resulting in artefacts. Fluorescent proteins and DNA paint also have blinking properties that can be quite different to ideal STORM data. HAWK is optimised for blinking ON times of a few frames. Supplementary Figs. 15, 16 show that HAWKMAN is still able to reliably identify artefacts for blinking rates well outside the optimum for HAWK. Results from HAWKMAN analysis are also not dependent on the choice of reconstruction pixel size (Supplementary Fig. 17).

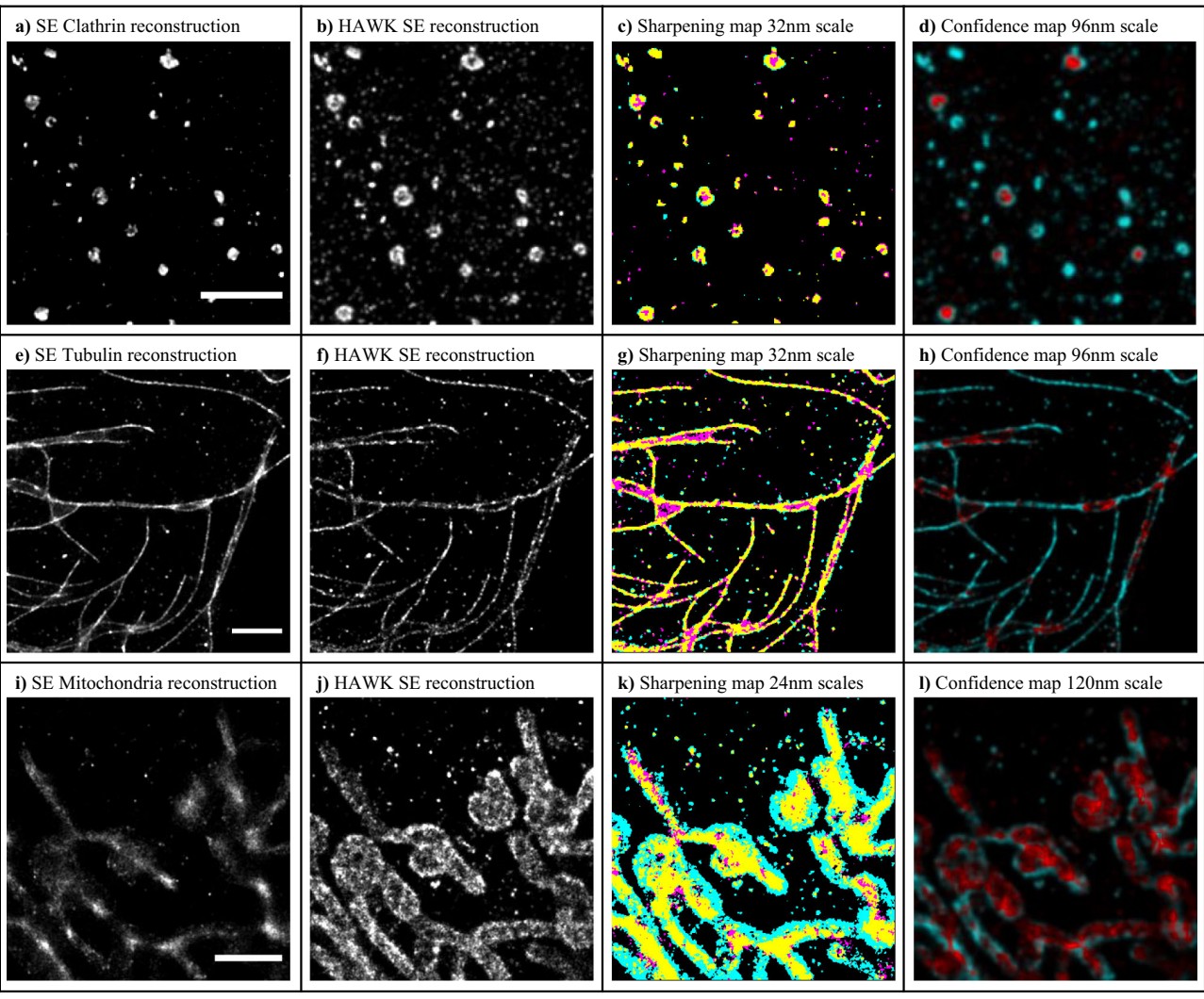

**Fig. 5 HAWKMAN can assess reconstructions from a variety of different structures.** Here, clathrin (**a–d**), tubulin (**e–h**) and the mitochondria epitope TOMM20 (**i–l**) have been imaged at medium-to-high density and evaluated with HAWKMAN. The test images were reconstructed using SE ThunderSTORM (**a**, **e**, **i**), and analysed with a reference image similarly reconstructed from the HAWK preprocessed dataset (**b**, **f**, **j**), and analysed using the sharpening map (**c**, **g**, **k**) and the confidence map (**d**, **h**, **l**). Data were displayed at a length scale of 32 nm (**c**, **g**) and 24 nm (**k**) for the sharpening maps, and at 96 nm (**g**, **h**) and 120 nm (**l**) for the confidence maps. The sharpening map threshold coefficients were 0.9 (**c**) and 0.7 (**g**, **k**). The confidence maps (**d**, **h**, **l**) highlight the substantial degrees of sharpening, particularly the TOMM20 reconstructions (**i–l**). Colouring as in Fig. 1, scale bars 2 μm.

The basis of HAWKMAN is that it exploits the fact that algorithms fail in different ways when used with and without HAWK. At extreme activation densities, most algorithms become significantly biased, whereas with HAWK there is no bias but a decrease in localisation precision. Therefore, when both images agree, one can be confident that the activation density is appropriate for that algorithm and the image contains the most accurate and precise localisations it can produce (i.e. comparable to a low density acquisition). That is to say, the HAWK (and test) reconstruction will not only be unbiased but also have a minimal precision loss.

Image artefacts are a critical issue in localisation microscopy. They can be very difficult to detect and quantify, even for experienced users, as they can involve subtle structure changes. In addition, they may only be present in certain regions of the image, with many parts of the image appearing of good quality. Therefore, providing tools which can detect artefacts below the resolution limit is critical if SMLM is to provide reliable and trustworthy results. HAWKMAN is a straightforward, fast (less than 1 min on a desktop PC for the challenge data presented in

Fig. 3) and reliable test, available as an ImageJ[18] plugin, which users can apply to verify the quality of their data.

## Methods

**HAWKMAN sharpening detection.** HAWKMAN analysis enables the similarity of two SMLM images to be assessed in a way which does not rely on the intensity values in the images being linearly related to the labelling density (or even each other), instead of concentrating only on structural differences, enabling us to reliably detect bias in the positions of structures. It should be noted that while this method will detect artefacts caused by errors in the SMLM data analysis, it will not detect artefacts which cause fluorophores to be imaged in a different location to the protein of interest (e.g. labelling artefacts). Therefore, to fully assess the authenticity of a super-resolution image, multiple methods of assessment may be necessary.

Two images are compared: the test image (which is undergoing quality assessment) and the reference image (which is produced using HAWK preprocessing and a fitting algorithm such as ThunderSTORM). For validating HAWKMAN, we have also sometimes used fits of low density data for the reference image, as a proxy for the ground truth. To avoid very bright points (from repeated localisations—frequently present in experimental data) leading to variable thresholding performance, we cap the maximum pixel intensity to the 98th percentile of the intensity histogram (not counting any zero valued pixels in the reconstruction).

A Gaussian blur is then applied to the image. The size (FWHM) of this blur determines the length scale at which the quality of the data is being assessed. By using multiple sizes of a blur, the local resolution of any part of the reconstruction (i.e. the length scale at which information starts to become locally unreliable) can be determined (see Supplementary Fig. 7). The algorithm will assess successive length scales (blur levels) up to a user-specified maximum, with each scale a progressively increasing integer (the scale No.) multiple of the reconstruction pixel size. The images are then normalised to a maximum intensity of one. From this starting point, three different mappings of errors are produced, each likely to highlight errors under different conditions.

The first assessment which we calculate is the sharpening map. An adaptive threshold for each image is calculated using Wellner's method[19]. Here we use a neighbourhood size $(2r + 1)$ of the largest odd number of pixels not greater than the current length scale. This produces a map of the mean intensity around each pixel. The images are then binarised based on whether the pixel intensity is above a set proportion $(C_b)$ of the local threshold $(P^{thr}_{x,y})$.

$$P^{thr}_{(x,y)} = C_b \frac{1}{(2r+1)^2} \sum_{x-r}^{x+r} \sum_{y-r}^{y+r} P_{(x,y)}, \; r = \text{ceil}\left(\frac{\text{scale No.}}{2}\right) - 1 \quad (1)$$

where ceil$(x)$ is the function that raises $x$ to the first integer greater than $x$. The value of $C_b$ determines what pixels in the image are classed as structure and which are classed as background. A value of $C_b = 0.7$ leads to a thresholded image where the width of linear structures is roughly equal to the FWHM of the intensity profile of a line profile through the structure in the original image. This value is designed to pick up features which have a peak-to-trough intensity ratio of about two or more. Where the image values between parts of a structure does not fall to this level, precise identification of the type of artefacts may require a higher threshold value, however, the default value will still detect that artefacts are present. This can be seen in the clathrin-coated pits imaged in Fig. 5 of the main text: here, the depth of the central hole is sometimes shallower than a factor of two, so a higher coefficient of 0.9 was used to also detect sharpening in these shallower pits. A guide to the optimum choice of threshold for structures of reduced contrast was produced by simulation of such structures and selecting the optimum coefficient, the results of which are shown in Supplementary Fig 3. This allows the optimum parameter value to be predicted by the user. For all simulations and experimental data from the Localisation Microscopy Challenge[15] and muscle sarcomeres, the standard coefficient of 0.7 was used.

At small blur scales (up to circa twice the localisation precision), local variations in labelling density and multiple reappearances of individual fluorophores can lead to falsely identifying these as very fine structures (microtubules are a common example of this). This can result from rapid increases in the adaptive threshold around these features, breaking the connectivity of the binarised structures. To counter this, we determined experimentally that an additional threshold of 0.1 times the calculated threshold at a neighbourhood size of half the PSF helped smooth out this false fine structure, by reducing the rate of spatial variation of the threshold. This stage is not strictly necessary to identify artefacts but gives a smother, better connected binarised image if this false fine structure is present. Additionally, a small baseline contribution to the threshold $(C_a = 0.04)$ eliminated many background localisations and reduced the fixed-pattern noise of some algorithms.

$$P^{thr}_{(x,y)} = C_b \frac{1}{(2r+1)^2} \sum_{x-r}^{x+r} \sum_{y-r}^{y+r} P_{(x,y)} + \frac{0.1}{(2r'+1)^2} \sum_{x-r'}^{x+r'} \sum_{y-r'}^{y+r'} P_{(x,y)} + C_a \quad (2)$$

$$r' = \text{ceil}\left(\frac{\text{PSF scale No.}}{2}\right) - 1$$

This noise threshold is designed to eliminate the background localisations that are not part of the labelled structure, with the magnitude of these in the reconstruction being assumed to be constant across the image. If it is not, regions of interest in each image can be taken and analysed separately using particularly noisy areas with a higher value for this coefficient. This was not necessary for any of the data analysed here. A colour overlay of the binarised images (the sharpening map) is then produced to reveal sharpening/bias artefacts. Areas, where there are substantial differences between the images, are indicative of sharpening in the test image or loss of precision in the reference. Areas, where the two images agree, can be considered reliable.

This 'sharpening map' is appropriate for detecting structure thinning and biases, but less suited to visualising the collapse of adjacent structures to one. The binarisation of the blurred and flattened images is repeated but with a higher threshold coefficient $C_b = 0.85$ (but never lower than the sharpening threshold if this is increased) and a baseline threshold of $C_a = 0.02$ to detect local maxima. These binarised images are then skeletonised using standard methods[18] giving a skeletonised interpretation of the structure. Again, a composite image of where the two structures agree and disagree reveals differences in structure collapse. Skeletonisation can lead to highly branched structures that are highly dependent on noise in the input images. This problem is greatly reduced by the initial Gaussian blur of the input images, particularly as the length scale increases. The skeletons produced obviously result in single pixel width structures that may only differ by single pixel displacements. This could possibly lead to a very high level of measured disagreement even at large length scales even though there are only differences of much smaller scale present in the images. We actually only want to consider differences that are on the scale of the current length scale. To achieve this the

skeletons are first re-blurred with a Gaussian blur kernel of FWHM equal to the current length scale. This ensures the overlap of the skeletons is substantial unless the differences in structure are comparable to or larger than the current length scale. The resulting comparison, the structure map, gives an indication of where in the sample and structures are biased to a greater degree than the length scale of interest.

The similarity between the test and reference output images is measured using the two-dimensional cross-correlation for both the sharpening and structure images (where the structure images are first Gaussian-blurred to the current length scale so as to become progressively tolerant of differences below this scale). A 'confidence map' is produced by summing the normalised, blurred test and reference images. This is then coloured according to the calculated local correlation, which indicates how likely the reconstruction is to be unbiased. Here, any local correlation above 0.85 in both the sharpening and skeletonised images is deemed as indicating valid structure. Areas below this level are coloured according to the level of agreement as measured by this correlation. The confidence level $S_{conf}$ is given by:

$$S_{conf} = \frac{1}{2} \min\left(1, \frac{PCC^{sharp}}{0.85}\right) + \frac{1}{2} \min\left(1, \frac{PCC^{str}}{0.85}\right) \quad (3)$$

where $PCC^{sharp}$ and $PCC^{str}$ are the Pearson correlation coefficients for the sharpening and structure comparisons respectively, calculated over a range equal to the current length scale, and $\min(x,y)$ indicates the smaller of $x$ or $y$. Due to the slightly reduced precision typical of the HAWK reference image and the effect of applying the Gaussian blur to the skeletons, a perfect correlation between test and reference images is unlikely even when the structures are very similar. Thus an 85% correlation threshold for full confidence is incorporated into the score in Eq. 3. This was determined as an appropriate value from all the simulated structures evaluated.

Similar to the Gaussian scale space framework[13], we consider this comparison at different length scales by convolving the input reconstructions with Gaussian kernels of different sizes. In this case, each Gaussian has an FWHM that is each integer multiple of the reconstruction pixel size, ranging from a single pixel up to the user specified maximum which is typically the instrument FWHM. This is suggested as the upper limit because typically this is the smallest separation at which most algorithms can regularly produce unbiased localisations. If the data are of very high density, even this scale may contain artefacts and if so, the calculation can be reperformed with a larger upper length scale limit. Gaussian scale space can be used to quantify the amount of information of a specific scale that is contained in an image. Here we are quantifying the difference in the structural information only between the test and reference reconstructions as a function of scale.

The above process is repeated for each length scale, producing an output image sequence wherein the degree of local sharpening is mapped as a function of scale. The actual reliable resolution obtained locally in the reconstruction can be assessed by observing at which scale disagreement between the test and the reference starts to occur.

The three maps described above provide a quantification of the degree of artefacts present at each length scale. If one wished simply to know which parts of an image are severely affected by artefacts, a summary of the results may be sufficient. As an optional step a map of the smallest length scale at which each part of the image has high confidence can be produced, the artefact scale map. For each pixel in the confidence map, the smallest length scale at which the confidence score is above 90% is recorded. Each pixel of the input image is then coloured according to this length scale with its intensity preserved. The image can then quickly be assessed to ascertain whether the features of interest are of sufficient authenticity.

**Experimental protocols: tubulin and clathrin.** Human cervical cancer cells (HeLa) were thawed and cultured in Dulbecco's Modified Eagle Media (DMEM) 1% Penicillin and Streptomycin (PS, Sigma-P0781), 10% foetal bovine serum (FBS, HyClone-SV30160.03) and 1% L-Glutamine (L-Glu, Sigma-59202C). Cells were transferred to a T25 flask (Cellstar-690175) and incubated at 37 °C and 5% Carbon dioxide. Passaging cells were trypsinized with 10% trypsin (Sigma-T4174) diluted in phosphate-buffered saline (PBS, Sigma-D8537), for 5 min once cells reached 80% confluence. Complete DMEM was added to neutralise trypsin and centrifuged at 960x$g$ for 3 min. The supernatant was aspirated, and cells resuspended in 5 ml of complete DMEM. Cells were plated on fibronectin (Sigma-FC010) coated 35 mm high glass bottom dishes (ibidi-81158) for imaging. Cells were fixed for 15 min in 3.6% formaldehyde (PeqLab-30201) at room temperature (RT) and washed three times with PBS.

Permeabilisation and blocking were undertaken by incubating cells in 'blocking buffer' consisting of 3% BSA (Sigma-10735108001) and 0.5% Triton X-100 (Sigma-X100) in $1 \times$ PBS for 10 min. Cells were then incubated with primary antibodies (tubulin from Sigma-T8328 at 1:200, clathrin from Abcam-ab21679 at 1:500) diluted in blocking buffer for 30 min while gently rocked at RT. After three 5 min washes in 'washing buffer' consisting in 0.2% BSA and 0.1% Triton X-100 in $1 \times$ PBS, cells were incubated with the Alexa Fluor® 647 secondary antibodies (Invitrogen-A21235 for tubulin, Invitrogen-A21244 for clathrin, both at 1:500 dilution) in blocking buffer for 30 min, gently rocked at RT. Cells underwent three 5 min washes in PBS×1 before being stored at 4 °C in $1 \times$ PBS for up to 2 days before imaging.

Cells were imaged in imaging buffer, where 1220 µl buffer was made using 800 µl distilled water, 200 µl 'MEA', 20 µl 'GLOX' and 200 µl 'dilution buffer'. Here, 'MEA' is 1 M cysteamine (Sigma-30070) and 0.25 M HCl (Sigma-H9892); 'GLOX' is 35 mM glucose oxidase (Sigma-G6125), 13.6 µM Catalase (Sigma-C40), 8 mM Tris (Amresco-E199) and 40 mM NaCl; lastly, 'dilution buffer' is 50 mM Tris, 10 mM NaCl (Alfa Aesar-A12313) and 10% w/v glucose (Thermo Fisher-G/0450). Dilution buffer was added just before imaging to minimise cell damage due to pH change.

Objective-based total internal reflection fluorescence (TIRF) was employed to minimise background. Fluorophore bleaching was undertaken with widefield illumination prior to image acquisition to minimise background signal from above the structures of interest. Approximately 5000 of a total 20,000 10 ms frames were acquired with supplementary 405 nm activation to obtain high density data.

Experimental data of clathrin and tubulin was gathered from fixed HeLa cells on a Nikon motorised inverted microscope ECLIPSE Ti2-E with Perfect Focus System in the King's College Nikon Imaging Centre. It is equipped with a laser bank with 405, 488, 561, and 640 nm lasers (LU-NV series), a 160-nm-pixel ORCA-Flash 4.0 sCMOS (scientific Complementary metal–oxide–semiconductor, Hamamatsu Photonics K.K.) and a CFI SR HP Apochromat TIRF 100XAC oil objective (NA 1.49) with an automatic correction collar. The microscope was controlled using Nikon Elements v5.2. software

**Experimental protocols: muscle sarcomeres.** Mouse cardiac myofibrils were prepared from freshly excised mouse cardiac muscle fibre bundles that had been tied to plastic supports to maintain an average sarcomere length of about 2.4 µm, as verified by laser diffraction. The fibre bundles were stored overnight at 0 °C in rigour buffer (140 mM KCl, 2 mM MgCl₂, 1 mM EGTA, 2 mM DTT, 20 mM HEPES, pH 6.8, containing protease inhibitors (Roche)). The next morning, the central sections of the fibres were dissociated by mechanical dissociation with a homogeniser following the protocol by (Knight, P.J. Meth. Enzymol., 1982. PMID: 7121291), washed in rigour buffer five times and stored in rigour buffer and at 0 °C until use. Suspensions of myofibrils in rigour-buffer were adhered to poly-lysine coated glass-bottomed dishes and fixed with 4% PFA in rigour buffer, washed in PBS and incubated in PBS/10% normal goat serum, before incubating with the rabbit polyclonal antibody Z1Z2 (binding to an epitope at the N-terminus of titin; Young, P. EMBO J., 1998. PMID: 9501083) diluted at 100:1, which labels the myofibrils at the Z-disc. Goat anti-rabbit secondary antibodies conjugated to Alexa647 (A-21244, Life Technologies, diluted 100:1) were applied after washing the samples in PBS, and visualisation performed after washing away unbound secondary antibody with PBS.

The apparatus used has been described in more detail in ref. [11]. Briefly, this consisted of a customised STORM microscope, built around a DMi8 Microscope body and 'SuMo' passively stabilised stage (Leica-microsystems GMBH). In this system, the 1.43 160X objective (Leica-microsystems GMBH) is mounted to the underside of the stage via a piezo drive (PI). The focus was maintained using a custom active control system. Excitation was from a 638 nm diode laser (Vortran). Emitter density was controlled with a 405 m Diode laser (Vortran). Fluorescence was collected in the 660–700 nm spectral range using dichroic filters (Croma). The imaging of sarcomere samples was performed in a standard reducing buffer (Glox-glucose,200 mM MEA). MicroManager v1.3 was used to acquire the images from the camera.

**Experimental protocols: mitochondria.** COS-7 cells (CRL-1651, ATCC) cultured on 25-mm-diameter coverslips (CSHP-No1.5-25, Bioscience Tools) were first fixed with 37 °C prewarmed 3% PFA (15710, Electron Microscopy Sciences) and 0.5% GA (16019, Electron Microscopy Sciences) in 1 × PBS (10010023, Gibco) at RT for 15 min. Subsequently, cells were washed twice with 1 × PBS and then quenched with freshly prepared 0.1% NaBH₄ (452882, Sigma-Aldrich) in 1 × PBS for 7 min. After washing three times with 1 × PBS, cells were treated with 3% BSA (001-000-162, Jackson ImmunoResearch) and 0.2% Triton X-100 (X100, Sigma-Aldrich) in 1 × PBS for 1 h, gently rocked at RT. Then cells were incubated with primary antibodies (sc-11415, Santa Cruz Biotechnology) at 4 °C overnight. After washing three times for 5 min each time with wash buffer (0.05% Triton X-100 in 1 × PBS), cells were incubated with secondary antibodies (A21245, Molecular Probes) at RT for 5 h. Both primary and secondary antibodies were diluted to 1:500 in 1% BSA and 0.2% Triton X-100 in 1 × PBS. After washing three times (5 min each time with wash buffer), cells were postfixed with 4% PFA in 1 × PBS for 10 min. Then cells were washed three times with 1 × PBS and stored in 1 × PBS at 4 °C until imaging.

Immediately before imaging, the coverslip with cells on top of it was placed on a custom-made holder. About 40 µL of imaging buffer (10% (w/v) glucose in 50 mM Tris (JT4109, Avantor), 50 mM NaCl (S271-500, Fisher Chemical), 10 mM MEA (M6500, Sigma-Aldrich), 50 mM BME (M3148, Sigma-Aldrich), 2 mM COT (138924, Sigma-Aldrich), 2.5 mM PCA (37580, Sigma-Aldrich) and 50 nM PCD (P8279, Sigma-Aldrich), pH 8.0) was added on top of the coverslip. Then another coverslip was placed on top of the imaging buffer. This coverslip sandwich was then sealed with two-component silicon dental glue (picodent twinsil speed 22, Dental-Produktions und Vertriebs GmbH).

SMLM imaging was performed on a custom-built setup on an Olympus IX-73 microscope stand (IX-73, Olympus America) equipped with a 100x/1.35-NA silicone-oil-immersion objective lens (FV-U2B714, Olympus America) and a

PIFOC objective positioner (ND72Z2LAQ, Physik Instrumente). Samples were excited by a 642 nm laser (2RU-VFL-P-2000-642-B1R, MPB Communications), which passed through an acousto-optic tuneable filter (AOTFnC-400.650-TN, AA Opto-electronic) for power modulation. The excitation light was focused on the pupil plane of the objective lens after passing through a filter cube holding a quadband dichroic mirror (Di03-R405/488/561/635-t1, Semrock). The fluorescent signal was magnified by relay lenses arranged in a 4f alignment to a final magnification of ~54 and then split with a 50/50 non-polarising beam splitter (BS016, Thorlabs). The split fluorescent signals were delivered by two mirrors onto a 90° specialty mirror (47-005, Edmund Optics), axially separated by 580 nm in the sample plane, and then projected on an sCMOS camera (Orca-Flash4.0v3, Hamamatsu) with an effective pixel size of 120 nm. A bandpass filter (FF01-731/137-25, Semrock) was placed just before the camera.

**Simulations.** Simulated data for parallel lines were taken from data used in the original evaluation of HAWK (see ref. [11] for detailed methods). Briefly, MATLAB was used to generate simulated microscopy image sequences using a Gaussian PSF of 270 nm and a camera pixel size of 100 nm. The labelling density was 100/µm along the lines. The active emitter density was controlled by varying the 'off' time. The number of frames was varied, so that in each case emitters make an average of five appearances regardless of the emitter density. Realistic photon and camera noise were included. Additional simulations were performed in the same manner for the 100 nm line separation used for Supplementary Fig. 3. Here additional emitters were positioned randomly between the lines in sufficient numbers to produce the required labelling density in the central region.

For the simulation of Z1Z2 antibody labelling of sarcomeres, these simulations were reperformed with the following modifications. Instead of each line consisting of a single column of fluorophores, three adjacent columns on the 10 nm spaced grid contained fluorophores, simulated a line of width 20 nm. The spacing between line centres was 60 nm resulting in an observable gap of 40 nm. The emitter density was controlled by varying the emitter-off time until the output closely resembled the experimental result. This corresponded to an activation density of 33 emitters/µm² of structure. Note this results in a high degree of emitter overlap in each frame than the same activation rate on the previous simulations due to the higher labelling density. The degree of sampling of each emitter was reduced to an average of 0.75 appearances per emitter, again this was adjusted for the best subjective match to experiment.

For the simulation of different blink statistics, the same approach was used except that the ON and OFF times and the number of frames were all varied by the same proportions to maintain the same emitter density and degree of sampling. For the 100 nm lines, three simulations were performed for ON times of 5, 15 and 25 frames. The activation density was equivalent to 10 emitters/µm² of structure. For the microtubules ON times of 1, 5 and 25 frames were performed at 33 emitters/µm² of structure. Equivalent low density simulations were performed at an activation density of 1 emitters/µm² of structure. All simulations used sufficient frames for each emitter to make an average of five blinks.

**Data analysis and image reconstruction.** All acquisitions underwent Thunder-STORM analysis as performed using the single and ME methods. Settings were as follows: Filter—Difference of Gaussians (min 1.0, max 1.6 pixels), Detector method —local maximum (connectivity 8), PSF model—Integrated Gaussian, Estimator—Maximum likelihood, fitting radius = 3. For ME fitting additional/alternate parameters were: fitting radius = 5, max molecules = 5, $p$ threshold = 0.05. SOFI analysis was used using the published MATLAB script [https://www.epfl.ch/labs/lben/lob/page-155720-en-html/page-155721-en-html/] which performs the calculation to fourth-order. In all cases, the balanced output was used. SRRF analysis was performed using the ImageJ plugin [https://doi.org/10.1038/ncomms12471] using the default parameters, except for where HAWKMAN was used to optimise some values, as described in Supplementary Fig. 6. Compressed sensing was performed using the published MATLAB script [https://doi.org/10.1038/nmeth.1978]. The model PSF was set to match that used in the simulations. DeconSTORM was also performed using MATLAB [http://zhuang.harvard.edu/decon_storm.html] with input parameters for fluorophore blinking were set to match the simulations. Due to computational limitations, the convolution method was used over the matrix method. Where HAWK preprocessing was used, identical parameters for the subsequent localisation step were used as in the case where HAWK was not used. The only additional processing was the filtering of false positive localisations by fitted width, following the procedure described in the original publication[11].

For the line pair simulations, image sequences were analysed using ThunderSTORM[12] (single and ME fitting), Balanced SOFI[17], SRRF[16], compressed sensing (CSSTORM)[20] and DeconSTORM[21]. For ThunderSTORM, SOFI and SRRF the full-size image of 64 × 64 pixels (100 nm size, 270 nm PSF) was analysed. Due to the large computational requirements of CSSTORM and DeconSTORM, the sequence was first cropped to 17 × 17 pixels for these methods. The reconstructions from the other algorithms were subsequently cropped to the equivalent area. Any offset between the different methods was adjusted at this stage, the required offset having been established by simulating a single point emitter. For all methods except SOFI, a magnification factor of 10 was used, giving a reconstruction pixel size of 10 nm. For SOFI, the maximum magnification factor

was 4, so these images were converted to 10 nm pixels size using bilinear interpolation (ImageJ).

Simulated data from the Localisation Microscopy Challenge were analysed in a near identical manner. For all analysis algorithms the same methods and settings were used as above, but making allowances for the differences in camera pixel size and gain. DeconSTORM was not used for this dataset, as the blinking properties of the simulated emitters were not known. Only ThunderSTORM was used on the low density simulation, as no significant emitter overlap should be present in this data.

Experimental Localisation Microscopy Challenge datasets were also analysed in a similar manner. For both SE and ME fitting with ThunderSTORM, all parameters were as above, except a magnification factor of 5 was used to render the images (corresponding to a 20 nm reconstruction pixel size). For SRRF with 'default parameters': ring radius = 0.3, Axes in ring = 6, Temporal analysis = TRPPM. For SRRF using 'optimum parameters': Ring radius = 0.3, Axis in ring = 8, Temporal analysis = TRAC order 2. In both cases the magnification factor was 5.

Experimental sarcomere data and all remaining simulated datasets were also analysed using ThunderSTORM SE fitting using the same parameters as above. Here the magnification was x10 (10 nm pixel size).

The Clathrin-coated pit, mitochondria and microtubule data were also processed in as similar way as possible, allowing only for differences between experimental setups. These data sets were all analysed using ThunderSTORM SE fitting, using the same parameters as the Localisation Microscopy Challenge data, with the exception of the differing camera pixel sizes these data sets were acquired on (160 nm for the microtubules and clathrin, 120 nm for TOM20). As the mitochondrial data was acquired on a bi-focal plane microscope, only a single plane was analysed.

**HAWKMAN analysis**. For all the analysis presented here, default threshold parameters were used, except for the clathrin-coated pit data, for which the thresholds were raised to 0.9 as described above and in the main text. The reconstructions using the default SRRF parameters on the Localisation Microscopy Challenge data included a low intensity fixed-pattern noise background of ~5% of the maximum intensity. To prevent analysing this as structure, this background was subtracted from the images before processing. This stage was not necessary for all other data.

**Statistics and reproducibility**. The results presented in this work do not contain or depend on any statistics drawn from multiple measurements. All the localisation algorithms used and the HAWKMAN analysis use computational methods that do not contain any random stochastic processes. Repetition of the analysis on the same input data should therefore always produce identical results. No recording or analysis of the repeatability of either the localisation algorithms, all of which have been published, or our HAWKMAN software have been performed. However, we observed no variation in output whilst testing the software on different computer systems

The test data taken from the Localisation Microscopy Challenge all consist of more than 1000 frames each of which contains a substantial number of overlapping emitters. The structures contain multiples areas of closely spaces structure where reconstruction bias would be expected in high density data. The reconstructions would therefore constitute the sum of a large number of individual measurements. These data sets are intended as a benchmark for assessing the performance of localisation analysis.

For the simulations performed here, the number of frames produced is adjusted so that emitters make an average of ca. 5 appearances regardless of density. Additionally, each structural element also contains at least ten emitters per camera pixel, this ensures a high degree of sampling and therefore repeatability of the associated reconstructions. The reconstructions produced from experimental data are provided as examples of use of the analysis on diverse biological systems. Each input reconstruction contains multiple incidences of identified artefacts, the precise nature of which will be sample-dependent. The repeatability of these experiments was therefore not examined but similar results were observed for the small number of other data sets examined.

The data presented in the plots and bar graphs consist of single calculated measured values resulting from a single experiment/simulation. No statistics were derived from these measurements which have no associated error. The line profiles from line pair data are averaged across the entire height of the image unless otherwise indicated in the figure (Supplementary Fig. 12).

**Ethical approval**. The use of animal tissue described in this paper was covered by the King's College London institutional license for tissue-only animal studies. No animal studies were performed in this work.

**Reporting Summary**. Further information on research design is available in the Nature Research Reporting Summary linked to this article.

## Data availability

The raw image sequences and reconstructions of original data supporting this research are downloadable from the Kings College London Research data server (https://doi.org/10.18742/RDM01-720). Raw data from the Localisation Microscopy Challenge are also

publicly available from http://bigwww.epfl.ch/smlm/index.html. Source Data is available as a source data file. Source data are provided with this paper.

## Code availability

The HAWKMAN analysis software in the form of an ImageJ plugin is provided as Supplementary Software.

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

## Acknowledgements

We thank D. Matthews for his assistance with the N-STORM system in the Nikon Imaging Centre at King's College London. This work was supported by the BBSRC (BB/R021767/1) and the MRC (MR/R008264/1). S.C. acknowledges support from a Royal Society University Research Fellowship and a Royal Society Enhancement award and I.C. acknowledges support from an EPSRC studentship.

## Author contributions

R.J.M. and S.C. conceived and designed the analysis and algorithm, with assistance from M.P. R.J.M. wrote the analysis software and produced simulations. All analysis of Localisation Microscopy Challenge data was performed by R.J.M. Preparation of samples and experimental measurements of microtubule and clathrin-coated pit data were performed by M.-A.G. and I.C. Mitochondria samples and data were prepared and collected

by D.M. and F.H. Analysis of experimental data was performed by I.C. aided by R.J.M. and S.C. Myofibril samples were prepared by M.G. and R.J.M. performed the experiments. The manuscript was produced by R.J.M., I.C. and S.C. All authors reviewed the manuscript.

## Competing interests

King's College London holds a patent on an analysis process (HAWK) used in portions of the research described in this manuscript (Authors R.J.M. and S.C. UK Patent Application No. 1800026.5) whose value may be affected by this research. The remaining authors declare no competing interests.
