## [Peer Review File · Nature Communications]

Reviewers' Comments:

Reviewer #1:

Remarks to the Author:

The paper by Marsh et al. addresses an open problem in the field of SMLM. In fact the issues has long been neglected by the community. With the arrival of ME fitting more attention to missing and incorrect localisations has been given. However, recent contributions like SQUIRREL have not made an impact in the community due to the fact that the method can only address diffraction limited image quality. As the authors point out it could in fact even falsely promote sharpening below the diffraction limit. Therefore I am happy to see this contribution that promises a (re)solution to this problem.

Overall I think the paper is well written and there is ample support for the conclusions in terms of exp. and simulation data. I have a few suggestions to be more clear in the wording in the text. On the methodical side, I have (many detail) questions and a number of suggestions that should simplify the algorithm. In addition detail is missing, i.e. I could not recode the algorithm given the information in the methods and SI. There are a number of things not considered in the text or SI that could impact the robustness of the algorithm. As main point I would suggest to remove the structure map from the measure for reasons given below.

Detail questions and suggestions:

* In the introduction: "Artificial sharpening occurs when emitters overlap in the raw data and are incorrectly localized towards their mutual centre, introducing a bias that is often substantial when compared with the estimated localization precision". Would that not result also in blurring depending on the geometry and dimensionality of the structure? Think about 2 close lines that would smear out?

* From the title, abstract, introduction it is a bit unclear what exactly the error-map (from the title) is. Then it seems the authors solely focus on "sharpening" from the abstract. Later the error is split into structure and sharpening. That could be streamlined, in particular in the beginning the focus is lost a bit here.

* "nonlinearity in the localisation imaging process". The authors never explicitly state what they mean by "non-linearity". The imaging process is linear of course in wide field, but the image processing part is non-linear by e.g. rejecting data etc.

* The authors make a case that non-uniform background is a problem for algorithms (top second page). I did not see a discuss on this or evaluation later with their method. The experimental data is collected in TIRF mode, essentially background free or not?

* The "degree of sampling" the authors mean the labelling density? Or the recovered density of localisations? Or ?

* With "intensity in the reconstruction" the authors mean the density of localizations, but they never say it. The localisations per se do not have an intensity, reconstruction algorithms do not put the photon count back into the image (and that is unreliable anyway).

* typo: "structures are authentically reproduce in the" p.1

* The authors should mention the use of Gaussian scale space in image processing since 1980s probably to asses what they have been doing with the blurring kernel in increasing sizes (old ref e.g. T. Lindeberg. Scale-space for discrete signals. IEEE Transactions on Pattern Analysis and Machine Intelligence, 12(3):234-254, 1990.). As this is an old field, the authors might find additional things they can use, for example how to choose the sampling of the blurring kernel.

* Relate is the question of what is the scale of the resolution/confidence? The author are much too vague on this. From the main text you also do not understand why so many different and not matching images with different scales are shown. This seems mainly hand picked. The SI movies show all of used scales, but what scale to pick is left unclear. I think the manuscript can gain here a lot.

In the methods part it reads "By using multiple sizes of blur, the local resolution (i.e. the length scale at which information starts to become locally unreliable) can be determined". "The actual

reliable resolution obtained locally in the reconstruction can be assessed by observing at which scale disagreement between the test and the reference starts to occur." How is this done? Is it done automatically? In S Fig 3,4,5 there are some picture, but what to choose? I think that could be done, as e.g. evolution of landmarks in scale-space is used to find relevant scale or look at the variance by scale etc. Here is a rich literature to explore.

* The use of skeletons and even more so the pixel-by-pixel overlap of skeletons from different images as done for the structure error gives me really a bad headache. Binary skeletons are not unique, not transferable to 3D in a reasonable way. The outcome depends typically on the end-point conditions, connectivity, but also on the *exact* implementation. Looking at the images the difference in the skeletons is only located at the junctions/crossings of the structures. I cannot expect that this can be robustly implemented at all or transferred. Skeletons are not used in IP to do pixel-by-pixel logical operations as done here, but for (initialisation of) segmentation and topological questions. In addition, skeletons are *very* noise sensitive (which again depends on the end-point conditions), therefore pixel-by-pixel comparison are tricky, really.

If you have a look at S Fig 3,8 you see that the skeleton is not stable across scales, which is expected but not a good property here.

In conclusion on the skeletonization, I think the authors should consider removing the structure map from the error-metric. They also more often do show only the other map.

* Figure 1 suggest that the structure map and the sharpening map are combined in the confidence. Is this is done by eq 3. If yes I would expect that PCC^{str} only has very little influence as it only is used where the skeleton is 1. Again a reason to remove this.

* Magic parameters:

It is nice that the authors explicitly state their parameters. In eq 2 C_b , 0.1, C_a and later 0.85, next to r . That is a handful. Next to C_a and r I think the others are really magic. There seems to be no reasoning other than we found they work for our test set of images. I think there could be some more consideration to them. In addition if other source code than for imageJ would be provided testing by others would be much easier.

The 0.85 and min operation in eq.3 make the confidence highly non-linear. Is that really desired?

* The super-resolution pixel size:

Is the algorithm dependent on the chosen reconstruction pixel size? A simulation study for one case would be sufficient to show this - and a guide how to choose it. Again for the skeletonization and the overlap operation (as this is a one pixel thick line) this could be a problematic point.

* "At small blur scales, local variations in labelling" in the methods. What is "small"? is this due to the fact that then locally the thresholding does not work?

* "suppressing intensity information" is too fancy a wording for thresholding or binarization, because that is what is done really. If this thresholding (with the same parameters) is successful depends a lot on the photo physics of the dye. How often does it come back etc? How do the experimental data compare in respect to this? I do not see PAINT data or PALM data with very much different statistics than in STORM. At least a simulation study would be nice to see here.

Bernd Rieger

Reviewer #2:

Remarks to the Author:

The paper by Marsh et al., describes an approach to map errors and confidence in super-resolution localization images by leveraging HAWK pre-processing. HAWK pre-processing has been established by the authors (March et al., Nature Methods 2018) and is proving a useful and reliable method to avoid and suppress the artifact of 'artificial sharpening' which can degrade super-resolution images. The trade-off is a small loss of localization precision performance and this paper provides a 'how to' procedure to identify areas of a super-resolution image that have had some loss of precision and therefore confidence.

Detailed workflows and explanations of all the steps are provided along with extensive testing and bench marking using a range of common localization algorithms and available test data. The results from this show – as does the original HAWK publication, and many others – that the artificial sharpening artefact arises from too high a spot density in the raw data. Treating this underlying problem is obviously the preferred approach to reducing artifacts however this is not always possible and therefore the use of data processing routines such as HAWK has a place. However, this manuscript does not contribute much more beyond the original HAWK publication other than providing 'how to' and validation of using HAWK to map where sharpening artifacts have been corrected by HAWK. Undoubtedly, what is in this manuscript will prove useful, and the paper is well written and everything in it properly established and it therefore should be published somewhere but it is not a Nature Communications paper. In short, there is no new scientific insight that requires urgent and prominent reporting. It is a confidence check and error mapping workflow utilizing an already established pre-processing routine. A more specialised journal would be appropriate.

Reviewer #3:

Remarks to the Author:

This paper shows us that all methods of super-resolution analysis can have errors with respect to sharpening and false structure predictions depending on the approach, parameter settings, and the chosen sample and that we cannot trust the image reconstruction wholly, to truly predict the structures with the resolution claimed by the method. This paper provides us with a tool to quantify the related errors and artifacts based on our initial analysis algorithm, by taking advantage of the method formerly published as HAWK.

The work is significant in the field since there is a need for quantification of errors in super-resolution microscopy and to do so in a unified manner for different analysis methods. The subject matter is of great importance since superresolution techniques are more and more used in biomedical research without proper knowledge of the limitations of analysis algorithms.

This is a solid paper, a derivative of the HAWK method [ref11] by the same group and comparable to FRC [ref8] and SQUIRREL [ref9] methods, with respect to the broad application for artifact investigation in super-resolution microscopy images (all three references were published in Nature Methods.)

The work has extensive data in the main article as well as supplementary information to showcase claims and conclusions and does that in a methodical step by step approach. Additional evidence to prove the claims is not necessary.

The methodology is sound and various aspects are investigated by the authors to showcase their conclusion in a satisfying manner. The work meets the expected standards of the field for sure and competes at the highest level. Methods provide enough details to reproduce the results.

There are no obvious flaws in the data analysis and conclusions and no major revision is needed. I enjoyed reading the paper. I hope it would be read by many researchers and that they understand the importance of it. I found the answer to most of my questions in the extensive supplementary section. I still have a few questions that I will put here. Most of them are around small yet, in my opinion, important technical details that would be helpful to the graduate student or the researcher at the microscope, if you add a couple of sentences about them. Also one or two questions that I found interesting about the possible application/limitation of HAWKMAN with respect to specific biological applications of super-resolution microscopy.

Question1: in super-resolution images, where the abundance of binding sites are enormously different on neighboring structures (for example, Actin bundles vs single filaments in the same vicinity) the variation in intensity creates a big difference in dynamic range, which causes the dimer structure not being detected. Can HAWKMAN resolve that issue since it's not centered around intensity measurements?

Question2: do these noise thresholds vary region by region in an image or are they implied over the whole image? If the thresholds are local, what are the sizes of regions normally picked for analysis? And what happens when two neighboring regions of different threshold are stitched up to make a whole image? In other words, since the local minimum can find different values based on the size of the region we pick for analysis (also a function of the box size in fitting codes), what are your thoughts on the effect of the ROI size on the confidence map?

Question3: related to the last question: if there are regions that are more crowded (or have higher background) and regions that are more sparse (with much lower background noise), does HAWKMAN pick the differences automatically and applies different thresholds? Or should the user imply that manually in the code (thus the need to divide the image into different regions and find local minimums per region to set the threshold)?

Question4: Would HAWKMAN treat non-specific binding different than the main structure? Is it possible to use confidence map to recognize and remove that artifact, since probably the sharpening pattern would look different than the main structure?

Question5: in the experimental setting, since the number of binding sites are limited, would a high-density area at a microtubule junction translate to high dye to antibody ratio as well? Would you see any difference in the sharpening between that and mere higher concentration of a one to one dye to antibody ratio?

Question6/Suggestion: Figure3, caption. "... at 3 pixels (60nm)..." do you mean your back-projected pixel size is 20 nm? That's not clear, written than way. Shouldn't it be about 100 nm based on the numbers we see later? Maybe add a few words in the caption to avoid probable confusion.

Question7: in Figure3, in a comparison between b and d, the question arises that, apparently if the background is noisy (which is not in these figures), the HAWK will pick that up and heighten it so it will increase the artifacts for smaller scales in the sharpening map. What is the relationship between noise and sharpening error? Again, does the size of the ROI that is picked for analysis effect this relation?

- based on the data presented and the discussion on two-color super-resolution, it seems that for example studies including co-localization with actin networks might benefit from the confidence map a lot, since many of those microdomains (open or closed) might be falsely visualized due to sharpening effects. Exciting.

Question8: in Figure 5b, we see a lot of background that is strengthened by HAWK. Does it mean that for small sparse binding sites (for example IgE receptors etc.), especially if not clustered or close-knit (so not like microtubules or Clathrin coated pits for example), making difference between the sharpened result and the ground truth would be difficult? What is your prediction in such cases? In other words, would sparsity and size put limitations in the performance reliability?

Finally, I strongly recommend publication of this paper based on the importance of this topic in biosciences and the extensive investigation of error-quantification in super-resolution analysis methods, by the authors.

Response to referees

We appreciate that the referees welcomed our new contribution to localisation microscopy data assessment, saying that ‘We are happy to see this contribution’ and commenting that the ‘work is significant in the field... The subject matter is of great importance’. Referee two does not consider allowing users to assess localisation microscopy data integrity to be an important subject, and so we must simply disagree (as do the other two referees). We note that referee two does not make any technical criticisms of our work.

We have made a number of changes to the paper based on the recommendations of referees one and three, and a detailed response to each referee is given below.

Referee 1: Bernd Rieger

** In the introduction: “Artificial sharpening occurs when emitters overlap in the raw data and are incorrectly localized towards their mutual centre, introducing a bias that is often substantial when compared with the estimated localization precision”. Would that not result also in blurring depending on the geometry and dimensionality of the structure? Think about 2 close lines that would smear out?*

We agree, and examples of this blurring are visible in the results presented. We had customarily referred to all artefacts caused by emitter overlap as ‘artificial sharpening’ as the bias in the localisation positions is towards their mutual centre. However, depending on the structure and how high the density of fluorophores is, this can lead to either sharpening or blurring of the structure. We have clarified this point throughout the manuscript and now describe the specific artefacts caused by biased localisations.

From the title, abstract, introduction it is a bit unclear what exactly the error-map (from the title) is. Then it seems the authors solely focus on “sharpening” from the abstract. Later the error is split into structure and sharpening. That could be streamlined, in particular in the beginning the focus is lost a bit here.

The referee is quite right here. While our method solely concentrates on errors produced by reconstruction algorithms that result from emitter overlap we do not distinguish this clearly from other types of error in the abstract and introduction. We have modified both to specifically refer to algorithm induced errors/bias introduced in high emitter density data. We also clarify what artefacts we refer to throughout rather than the generic term ‘artificial sharpening’ - see specific points below for details.

** “ nonlinearity in the localisation imaging process “. The authors never explicitly state what they mean by “non-linearity”. The imaging process is linear of course*

in wide field, but the image processing part is non-linear by e.g. rejecting data etc.

The referee is right to point out the lack of clarity. The pixel intensities in a localisation microscopy reconstruction ideally do not reflect the emitter photon intensities but the number density of localisations. We use the term here as it is a central assumption of the SQUIRREL image assessment method that pixel values in the reconstruction are linearly related to the intensity in the widefield image. This will not be true when the emitter density rises as all localisation algorithms miss localisations and therefore undercount the number of emitters – breaking its linear relationship. We have expanded our description of this relationship in the relevant sections along the lines suggested (top page 2, bottom page 8 & top page 9).

Also we have clarified our use of the term ‘localisation imaging process’ to refer specifically to the process of localising emitters and producing a reconstruction (the image) to distinguish it for the separate process of acquiring the sequence of widefield images from the camera, which is not considered in the HAWKMAN method. We thank the referee for pointing out this ambiguity.

** The authors make a case that non-uniform background is a problem for algorithms (top second page). I did not see a discuss on this or evaluation later with their method. The experimental data is collected in TIRF mode, essentially background free or not?*

The central point of our discussion on non-uniform background is how it can contribute to SQUIRREL incorrectly ranking reconstructions in certain circumstances. In our experience in using it this can have significant effects even when quite small. We have expanded our description of how background fluorescence may affect SQUIRREL in the section on non-linearity mention above. The experiments were performed in TIRF but for the structures studied still contained sufficient background that might pose a problem for SQUIRREL. How variations in background of scale comparable to the separation of adjacent emitters effects bias in their localisation in high density data is a complex and under explored subject, which is likely to be quite different for different localisation methods (Gaussian fitting vs SRRF for example). We only mention it as a possible cause of localisation bias. We have therefore removed the comment on the subject as it does not contribute anything to or constitute any part of the method.

** The “degree of sampling” the authors mean the labelling density? Or the recovered density of localisations? Or ?*

Apologies for the lack of clarity here: “by degree of sampling” we mean the variation in the proportion of labels that have made an appearance and been successfully localised. We discuss this point specifically in relation to SQUIRREL, which requires that the pixel intensity in the reconstruction be a direct

representation of the labelling density. In the wide-field image it can be reasonably assumed that all emitters make an equal contribution to the signal; for the reconstructed image, some emitters may be localised several times while others may not be at all. In high emitter density situations this tendency may correlate with the structure due to counting overlapping emitters as a single localisation. This prevents the reconstruction being a simple linear transform of the widefield image. We have expanded and clarified our description at the appropriate points in text (top page 2 and throughout)

** With "intensity in the reconstruction" the authors mean the density of localizations, but they never say it. The localisations per se do not have an intensity, reconstruction algorithms do not put the photon count back into the image (and that is unreliable anyway).*

By intensity in the reconstruction we mean the pixel values in the input reconstructed image. For localisation microscopy that is related to the density of localisations depending on the exact method used to display the data (histogram, Gaussian blur etc). Other methods such as SOFI do have some information related to the photon intensities in their reconstructions. We make no assumptions about the display method used to make our algorithm as generally applicable as possible. We accept the ambiguity of our language on this matter and have change the manuscript throughout to be more specific.

** typo: "structures are authentically reproduce in the" p.1*

Corrected

** The authors should mention the use of Gaussian scale space in image processing since 1980s probably to assess what they have been doing with the blurring kernel in increasing sizes (old ref e.g. T. Lindeberg. Scale-space for discrete signals. IEEE Transactions on Pattern Analysis and Machine Intelligence, 12(3):234-254, 1990.). As this is an old field, the authors might find additional things they can use, for example how to choose the sampling of the blurring kernel.*

We thank the reviewer for drawing our attention to the work on Gaussian scale space. Inspired by this, we have extended our method to quantify the length scale at which artefacts are not significant. This 'Artefact scale map' gives information similar to resolution, but for bias.

We have added a discussion of the similarities with Gaussian scale space (page 4) and a description of how the way the Artefact scale map is calculated. It is located at the end of the section of the main paper that describes the algorithm with reference to the flow chart in figure 1 (page 4). A further discussion of how Gaussian phase space inspired the Artefact scale map , and more detail of its calculation is found at the end of the first part of the method section (page 16).

** Relate is the question of what is the scale of the resolution/confidence? The authors are much too vague on this. From the main text you also do not understand why so many different and not matching images with different scales are shown. This seems mainly hand picked. The SI movies show all of used scales, but what scale to pick is left unclear. I think the manuscript can gain here a lot.*

The images presented in the figures are for length scales that were chosen simply to provide examples of the HAWKMAN output. There is nothing specific or important about the length scales chosen. We simply wished to show that there are different criteria that HAWKMAN uses to assess quality, the three maps, and that these produce information at all length scales. They were simply chosen to span the range of length scales analysed in a single figure. When comparing performance, we always show the same length scales for each algorithm.

Each length scale gives an assessment in its own right and there is no need to select any relevant length scale. The intent is simply to map where errors of a specific size occur in the reconstruction, i.e. for errors of scale 50nm use the 50nm length scale. The 200nm length scale results would map errors of up to 200nm in size. To aid in the interpretation and presentation of the HAWKMAN results, we have adapted our algorithm to additionally produce a single frame map of what is the largest scale artefact at each point in the reconstruction (the artefact scale map). This is generated from the confidence maps and indicates the smallest length scale that HAWKMAN reports high confidence in for that part of the reconstruction. We have also clarified these points in the text (pages 6, 15 & throughout), and hope that this clarifies the issue and the referee feels this will aid users in interpreting the HAWKMAN output.

The description of how this map is calculated and its intended use can be found in the section that describes the HAWKMAN algorithm (using figure 1 flow chart) and in more detail in the methods section (page 16). Results from the map have been shown throughout the paper where space in the figure allows.

In the methods part it reads "By using multiple sizes of blur, the local resolution (i.e. the length scale at which information starts to become locally unreliable) can be determined". "The actual reliable resolution obtained locally in the reconstruction can be assessed by observing at which scale disagreement between the test and the reference starts to occur." How is this done? Is it done automatically? In S Fig 3, 4, 5 there are some picture, but what to choose? I think that could be done, as e.g. evolution of landmarks in scale-space is used to find relevant scale or look at the variance by scale etc. Here is a rich literature to explore.

We refer the referee to the statements and changes made in response to the previous comment about this combined subject and hope this, along with the addition of the artefact scale map analysis, provides sufficient explanation and clarification.

** The use of skeletons and even more so the pixel-by-pixel overlap of skeletons from different images as done for the structure error gives me really a bad headache. Binary skeletons are not unique, not transferable to 3D in a reasonable way. The outcome depends typically on the end-point conditions, connectivity, but also on the *exact* implementation. Looking at the images the difference in the skeletons is only located at the junctions/crossings of the structures. I cannot expect that this can be robustly implemented at all or transferred. Skeletons are not used in IP to do pixel-by-pixel logical operations as done here, but for (initialisation of) segmentation and topological questions. In addition, skeletons are *very* noise sensitive (which again depends on the end-point conditions), therefore pixel-by-pixel comparison are tricky, really. If you have a look at S Fig 3,8 you see that the skeleton is not stable across scales, which is expected but not a good property here. In conclusion on the skeletonization, I think the authors should consider removing the structure map from the error-metric. They also more often do show only the other map.*

The referee highlights several concerns over the use of skeletonization of the binarized images, specifically the pixel by pixel comparison of the test and reference skeletons. From these comments we wonder if we have not been clear about one of the steps in the process. Before any quantitative comparison is made the skeletons are re-blurred. This is done by convolution with a Gaussian kernel of FWHM equivalent to the current length scale used in each iteration of the algorithm. This is the same kernel used to blur the input images. The correlation is then calculated using these blurred images. This has the effect of smoothing out the instabilities and noise sensitivity in the skeleton that the referee mentions, particularly as the length scale increases. We display the skeletons before this blurring stage which may well have contributed to a lack of clarity: we have therefore changed all the figures to display the post-blur skeletons, and have made this the default method of display in the plugin.

The reason for inclusion of the skeleton is that it (in combination with the higher relative threshold for binarization) is more sensitive to artefacts than the sharpening metric in certain situations, particularly when there is structure that is only just resolvable in the HAWK reference. Density-induced localisation bias can result in a shift in both the mean and peak positions of the transverse profile of a structure. The relative amounts of these distortions can vary considerably with the structure and degree of emitter overlap. The skeleton is aimed at detecting shifts in the peak of the profile while the sharpening map is more sensitive to the mean position and width of the profile. For the microtubule and sarcomere data presented in the supplementary information, we include a plot of the relative contributions of the sharpening and structure maps to the overall correlation score for the whole image. These show that the structure map can show much less, or indeed more agreement than the sharpening map depending on the length scale, the structure, the density and the algorithm used. For these reasons we would argue that the structure map provides a useful contribution to identifying artefacts. However, we concede that this distinction and reasoning is not made very clear in the manuscript and have expanded the appropriate sections to explain why we use both methods of assessment and describe their relative merits (pages 15, 16 & throughout).

** Figure 1 suggest that the structure map and the sharpening map are combined in the confidence. Is this is done by eq 3. If yes I would expect that PCC^{str} only has very little influence as it only is used where the skeleton is 1. Again a reason to remove this.*

The referee points out that the correlation score for the structure and sharpening maps are combined in equation 3 but questions whether the nonlinear nature of it is desirable. The range over which the correlation is calculated at each pixel location is equivalent to the current length scale and the images used for PCC^{str} are the re-blurred skeletons described above. This means the calculated correlation drops off steadily, rather than abruptly, as the disagreement between skeletonised structures increases. The length scale specific blurring means the correlation falls more slowly with distance as the length scale is increased resulting in a greater tolerance or error for larger length scales.

The inclusion of the 85% confidence threshold in equation 3 does introduce non-linearity to the 'score'. This is necessary to account for the differing nature of the HAWKed reconstructions to their no-HAWK counterparts. The HAWK reconstruction generally will have lower localisation precision than the test image, an effect that increases as the emitter overlap increases. Additionally, the relationship between emitter density and reconstruction pixel intensity is different in the HAWK image as more emitters are detected and may be counted repeatedly. An exact match between images is therefore unlikely and the threshold of 85% correlation for a perfect score prevents correct structure from frequently being detected as containing small artefacts. This value was optimised from the analysis of simulated high and low-density data. We added a clearer description of the purpose of this threshold to the methods section (page 15).

** Magic parameters:*

It is nice that the authors explicitly state their parameters. In eq 2 C_b , 0.1, C_a and later 0.85, next to r . That is a handful. Next to C_a and r I think the others are really magic. There seems to be no reasoning other than we found they work for our test set of images. I think there could be some more consideration to them. In addition if other source code than for imageJ would be provided testing by others would be much easier.

We agree with the referee's dislike of adjustable parameters, particularly if their purpose, values and effect on the analysis is not intuitive. In this and the HAWK algorithm we have strived to minimise the number of parameters. However, to keep the process applicable to differing types of structure and super-resolution processes other than just Gaussian fitting, some parameterisation was necessary. The purpose of the C_b coefficient is to determine how close to the local maximum the adaptive threshold for binarization is set. The default value of 0.7 produces a threshold of ca. 50% of the local maximum when the tangential profile of the structure in the reconstruction is approximately Gaussian. For some

2D structures which contain areas of differing labelling density the profiles can be highly non-symmetrical or contain a plateau. If this is sufficiently high, the standard threshold will not discriminate between them, counting both as structure. If it is wished to distinguish the former from the latter a higher threshold (higher value of C_b) is needed so that the lower density region is not counted as structure. The clathrin-coated pit images in the main text are an example of this, where the reconstructed density inside the pits is typically in the range 50% - 75% of the perimeter. The threshold coefficient was raised in this example to distinguish the parameters in the binarized images. To aid in the selection of an increased threshold in these circumstances we have used simulation to produce a calibration curve for this parameter given the relative labelling densities. The results are presented in an additional supplementary figure (Supp. Fig 3). Adjustment of this parameter is only necessary to distinguish the perimeter structure from the central structure in the HAWK reference and therefore identify the 'infilling' of this hole as the type of artefact. Without adjustment HAWKMAN will still detect the presence of artefacts as the size of the overall structures will still differ between the test reconstruction and the HAWK reference.

HAWKMAN was implemented ImageJ plugin to facilitate the easy and wide uptake. Full versions of the source for the ImageJ plugin (used for all the results presented in the manuscript) for both HAWK and HAWKMAN is available on request. A developmental version of the HAWKMAN method was implemented in MATLAB that does not contain all the features or review revisions of the plugin but does describe the basic method. This version could be adapted with appropriate comments and pseudo code and made available on request.

** The super-resolution pixel size:*

Is the algorithm dependent on the chosen reconstruction pixel size? A simulation study for one case would be sufficient to show this - and a guide how to choose it. Again for the skeletonization and the overlap operation (as this is a one pixel thick line) this could be a problematic point.

The referee is concerned that the results may be dependent on the super resolution pixels size. We wonder if this is because of the confusion over the re-blurring of the skeletons mentioned above. As the blur is proportional to the length scale, blurred skeletons at the same absolute length scale should be virtually independent of pixels size. As suggested, we have created a supplementary figure (Supp fig 17) comparing results using the Localisation Microscopy challenge data. These show results of HAWKMAN analysis are very similar for a wide range of super resolution pixel sizes.

"At small blur scales, local variations in labelling" in the methods. What is "small"? is this due to the fact that then locally the thresholding does not work?

Yes, we should have specified which length scales we consider small. Here we mean scales up to about twice the localisation precision. Reconstructions of long filamentous objects of width comparable to the localisation precision or less (such as microtubules or actin fibres) frequently contain clusters of localisations

that give the appearance of fine spot like structures superimposed on top of the filament structure. These may arise from antibody clustering or frequently appearing emitters. The scale of these clusters will appear as close to the localisation precision. A single, short range, adaptive threshold may break the connectivity of the binarized structure where these features are of significant magnitude, as the adaptive threshold level is raised rapidly in the vicinity of these features. When the length scale is increased, the Gaussian blur tends to smooth out these variations and the single adaptive threshold successfully identifies the underlying structure. We therefor added a component to the threshold calculation that is calculated over a longer range, reducing the rapid rise of the threshold near these features resulting in a better connected binarized structure. We have added this explanation to the methods section (page 15).

** "suppressing intensity information" is too fancy a wording for thresholding or binarization, because that is what is done really. If this thresholding (with the same parameters) is successful depends a lot on the photo physics of the dye. How often does it come back etc? How do the experimental data compare in respect to this? I do not see PAINT data or PALM data with very much different statistics than in STORM. At least a simulation study would be nice to see here.*

Hopefully we have clarified what we mean by intensity information in super-resolution reconstruction in our earlier responses. Our intention was to convey that we wish to restrict our comparison to basic structural information and remove as much as possible any dependence on the consequences of varying sampling and detection rates of different algorithms. We have altered the language throughout the text to better reflect that it is indeed a process of thresholding and binarization.

It is certainly true that the blinking statistics can greatly influence the quality and resolution of a reconstruction. The use of binarized structures in performing the comparison is also intended to greatly reduce the influence of blinking statistics, which may have different effects on the reconstructions produced different algorithms. This is why we only make comparisons with a reference that uses the same localisation algorithm, produced from the same initial data. HAWKMAN was designed to detect faults that result from excess emitter density and assumes that any other errors resulting from experimental factors, once binarized, will be equal in the test and reference images. One exception to this could be the number sequential frames an emitter appears in - the ON time. HAWK works distinguishing emitters by their differing on times. Large changes in this parameter could have detrimental effect on the performance of HAWK. We have therefor performed simulation studies on a range of ON times, including those better reflecting the PALM/DNA paint methods described. We have added supplementary figures (supp fig 15 * 16) demonstrating that HAWKMAN still performs well in these circumstances and have made reference to these in the main text.

Reviewer #2 (Remarks to the Author):

The paper by Marsh et al., describes an approach to map errors and confidence in super-resolution localization images by leveraging HAWK pre-processing. HAWK pre-processing has been established by the authors (March et al., Nature Methods 2018) and is proving a useful and reliable method to avoid and suppress the artefact of 'artificial sharpening' which can degrade super-resolution images. The trade-off is a small loss of localization precision performance and this paper provides a 'how to' procedure to identify areas of a super-resolution image that have had some loss of precision and therefore confidence.

Detailed workflows and explanations of all the steps are provided along with extensive testing and bench marking using a range of common localization algorithms and available test data. The results from this show - as does the original HAWK publication, and many others - that the artificial sharpening artefact arises from too high a spot density in the raw data. Treating this underlying problem is obviously the preferred approach to reducing artefacts however this is not always possible and therefore the use of data processing routines such as HAWK has a place.

However, this manuscript does not contribute much more beyond the original HAWK publication other than providing 'how to' and validation of using HAWK to map where sharpening artefacts have been corrected by HAWK. Undoubtedly, what is in this manuscript will prove useful, and the paper is well written and everything in it properly established and it therefore should be published somewhere but it is not a Nature Communications paper. In short, there is no new scientific insight that requires urgent and prominent reporting. It is a confidence check and error mapping workflow utilizing an already established pre-processing routine. A more specialised journal would be appropriate.

The ability to produce verifiable and correct results is generally considered critical in science. If only new scientific insight is given prominent reporting, we run the great risk of reporting incorrect and artifactual results. We would argue that novel methods of ensuring that correct results are produced underpin the whole of the scientific endeavour, and should be brought to the attention of those working in the field. This is clearly a view held by others since, for example, the method SQUIRREL, an error detection method for super-resolution images which cannot detect sharpening artifacts or errors below the standard resolution limit, was published in Nature Methods. Similarly Fourier Ring correlation, a commonly used resolution measure, was published in the same journal.

Our method is the only one able to detect sharpening artifacts significantly below the diffraction limit, the only one able to detect when the reconstruction may contain unresolved structure due to localisation bias, and the only one able to produce a resolution scale at which the image is artifact free. We are therefore providing a method which solves problems present in the two most popular localisation data assessment tools (SQUIRREL and FRC), which could cause people to substantially overestimate the quality of their data. Previous precedent

clearly indicates this will be a widely used and popular technique, and we therefore think it is very suitable for Nature Communications.

Referee 3: Farzin Farzam

Referee 3 also makes highly complimentary comments, noting both the manuscripts quality and relevance/impact. They suggest no major revision but has a number of detail/application questions. We greatly appreciate these comments and will attempt to answer them.

Question1: in super-resolution images, where the abundance of binding sites are enormously different on neighboring structures (for example, Actin bundles vs single filaments in the same vicinity) the variation in intensity creates a big difference in dynamic range, which causes the dimer structure not being detected. Can HAWKMAN resolve that issue since it's not centered around intensity measurements?

The referee asked about adjacent structures of very different labelling density. Sharpening artefacts tend to be more extreme in these circumstances. At high emission density the appearance of an emitter attached to the infrequently labelled structure may always be accompanied by the simultaneous appearance of one or more emitters from the densely labelled structure. Almost all localisations of the sparsely labelled structure would be biased or missed, resulting in the reduced dynamic range the referee speaks of. As HAWK detects intensity fluctuations it is relatively unaffected by this effect. We also refer the referee to our comments on the threshold parameters in response to referee one above.

Question2: do these noise thresholds vary region by region in an image or are they implied over the whole image? If the thresholds are local, what are the sizes of regions normally picked for analysis? And what happens when two neighbouring regions of different threshold are stitched up to make a whole image? In other words, since the local minimum can find different values based on the size of the region we pick for analysis (also a function of the box size in fitting codes), what are your thoughts on the effect of the ROI size on the confidence map?

The basic noise threshold C_a is constant throughout the image. For where there are areas of the input images with very high noise and background, it may be beneficial to divide the images into segments and increase the noise threshold for these regions. There should be no issues with stitching the outputs back together so long as the size of the regions is large compared to the longest length scale used. The distinction between sparse and densely labelled regions is related to the second threshold coefficient and we refer the referee to the comments made on this subject in response to referee one above and the new supplementary figure (Supp fig 3)

Question3: related to the last question: if there are regions that are more crowded (or have higher background) and regions that are more sparse (with much lower background noise), does HAWKMAN pick the differences automatically and applies different thresholds? Or should the user imply that manually in the code (thus the need to divide the image into different regions and find local minimums per region to set the threshold)?

We refer the referee to the comments made to the previous questions and hope that this answers the question satisfactorily.

Question4: Would HAWKMAN treat non-specific binding different than the main structure? Is it possible to use confidence map to recognize and remove that artefact, since probably the sharpening pattern would look different than the main structure?

Non-specific binding would be treated in a similar way to the less densely labelled central regions in the clathrin pit data given. It is an example of when the threshold coefficient may need to be increased, as described above. We have added this as an example in the methods section. If non-specific labelling is present at sufficient density to cause localisation bias, then HAWKMAN would be expected to detect it as such. However, it will not reliably distinguish this as being a different kind of artefact from the directly labelled structure.

Question5: in the experimental setting, since the number of binding sites are limited, would a high-density area at a microtubule junction translate to high dye to antibody ratio as well? Would you see any difference in the sharpening between that and mere higher concentration of a one to one dye to antibody ratio?

Whilst in localisation microscopy you are always imaging the positions of emitters not antibodies or the targeted protein, it can be challenging to get sufficient resolution for these differences to be relevant even in low emitter density data. For high density data the localisation performance of even of specialist algorithms is substantially reduced (see for example the Localisation Microscopy Challenge). We don't know of any algorithm that would produce a different amount of bias for two simultaneous emitters bound to the same antibody or to adjacent ones unless the separation was a few tens of nanometers or more.

Question6/Suggestion: Figure3, caption. "... at 3 pixels (60nm)..." do you mean your back-projected pixel size is 20 nm? That's not clear, written than way. Shouldn't it be about 100 nm based on the numbers we see later? Maybe add a few words in the caption to avoid probable confusion.

Yes, we do mean back projected/reconstruction pixel size. We always refer to this throughout the manuscript when we mention pixel size. The reason the sharpening map is shown at 60nm and the confidence map is shown at 100nm was to give the maximum range of examples for the full HAWKMAN output in the limited space of a figure. We have modified the captions and manuscript to make this clearer as suggested.

Question 7: in Figure 3, in a comparison between b and d, the question arises that, apparently if the background is noisy (which is not in these figures), the HAWK will pick that up and heighten it so it will increase the artefacts for smaller scales in the sharpening map. What is the relationship between noise and sharpening error? Again, does the size of the ROI that is picked for analysis effect this relation?

- based on the data presented and the discussion on two-color super-resolution, it seems that for example studies including co-localization with actin networks might benefit from the confidence map a lot, since many of those microdomains (open or closed) might be falsely visualized due to sharpening effects. Exciting.

Yes, if the acquisition background is very noisy then HAWK may pick up false localisations that are not present in the test reconstruction. In this case HAWKMAN would identify these localisations as very low confidence. This is also true for very low intensity slow blinking emitters that are too dim to be detected in the un-HAWKed sequence but are picked up in the longer filters of HAWK, which takes contributions from several frames. If these emitters are bound to the protein and are substantial in number, HAWKMAN would incorrectly associate these with an artefact. However, in our experience these much more frequently from unbound background emitters. The ROI may have a modest effect on the number of these localisations identified, as the base noise threshold C_a is relative to the highest intensity pixel in the intensity-flattened image. We have expanded on the description of appropriate coefficient values in the methods section and added a note on using ROIs (page 15).

Question 8: in Figure 5b, we see a lot of background that is strengthened by HAWK. Does it mean that for small sparse binding sites (for example IgE receptors etc.), especially if not clustered or close-knit (so not like microtubules or Clathrin coated pits for example), making difference between the sharpened result and the ground truth would be difficult? What is your prediction in such cases? In other words, would sparsity and size put limitations in the performance reliability?

Yes, if the labelling of a structure is quite sparse and therefore the spatial sampling of the structure quite low, a comparison with the ground truth would become more difficult for the HAWK image if it also contained a lot of background localisations resulting from noise. However, the differences in bias of the two input images should remain unaffected. HAWKMAN should therefore still be able to identify if the reconstruction contains artefacts resulting from excess

emitter density even though the HAWK reconstruction may be of fairly low quality. The additional noise localisation in HAWK can be hard to distinguish from very small sparsely labelled structures like IgE receptors. These features would be absent from the test reconstruction however and would be marked as low confidence. HAWK can often detect very low intensity, slow blinking emitters that were missed in the test reconstruction that do correspond to real structure. In These circumstances HAWKMAN may falsely identify some of the true structures present in the HAWK image but missing from the original as artefacts. However, a false positive detection of an artefact would be much preferred to a false negative.

Reviewers' Comments:

Reviewer #1:

Remarks to the Author:

The authors made a great effort with the revision. All my comments and/or concerns have been resolved. Some were misunderstandings from my side and the authors added/changed text to clarify it. Other comments have been taken to heart and incorporated as such.

My most critical technical comment on the not robust skeletonization has been resolved.

I am very happy that the "artefact scale map" was introduced. To me this one image directly shows me where to trust a reconstruction and where not. As a suggestion the authors might want to change the lookup scale bar for the colours from pixel units to physical units in the subfigures 2m,n,o) and then later too.

Reviewer #3:

Remarks to the Author:

I'm satisfied with the answers provided to my questions and the changes made based on the suggestions. I think it is of importance for the readers of this manuscript to have access to the questions asked by the reviewers and the answers given, especially for those group of readers who are the end users of these techniques and are not endowed to go deep into the technical aspects of the algorithm and the method but will benefit from using the technique in answering biological questions and designing relevant experiments. At least the part of questions that are application related would help some of my colleagues and their students immensely in finding about the advantages and limitations of the technique, so I encourage the author(s) to consider publishing reviewer comments and the author responses. I'm very happy with the status of this paper and the changes made based on comments made by all reviewers. Kudos.

Response to referees comments.

Reviewer #1 (Remarks to the Author):

The authors made a great effort with the revision. All my comments and/or concerns have been resolved. Some were misunderstandings from my side and the authors added/changed text to clarify it. Other comments have been taken to heart and incorporated as such.

My most critical technical comment on the not robust skeletonization has been resolved.

I am very happy that the "artefact scale map" was introduced. To me this one image directly shows me where to trust a reconstruction and where not. As a suggestion the authors might want to change the lookup scale bar for the colours from pixel units to physical units in the subfigures 2m,n,o) and then later too.

We thank the reviewer for their supportive comments about our revisions. We have taken on board their suggestions about the colour scale labels and have modified the figures accordingly.

Reviewer #3 (Remarks to the Author):

I'm satisfied with the answers provided to my questions and the changes made based on the suggestions. I think it is of importance for the readers of this manuscript to have access to the questions asked by the reviewers and the answers given, especially for those group of readers who are the end users of these techniques and are not endowed to go deep into the technical aspects of the algorithm and the method but will benefit from using the technique in answering biological questions and designing relevant experiments. At least the part of questions that are application related would help some of my colleagues and their students immensely in finding about the advantages and limitations of the technique, so I encourage the author(s) to consider publishing reviewer comments and the author responses. I'm very happy with the status of this paper and the changes made based on comments made by all reviewers. Kudos.

Again we thank the reviewer for their supportive comments about our revisions.